# GS-CPR: Efficient Camera Pose Refinement via 3D Gaussian Splatting

**Changkun Liu**[1][*]   **Shuai Chen**[2]   **Yash Bhalgat**[2]   **Siyan Hu**[1]   **Ming Cheng**[3]
**Zirui Wang**[2]   **Victor Adrian Prisacariu**[2]   **Tristan Braud**[1]
[1]HKUST   [2]University of Oxford   [3]Dartmouth College

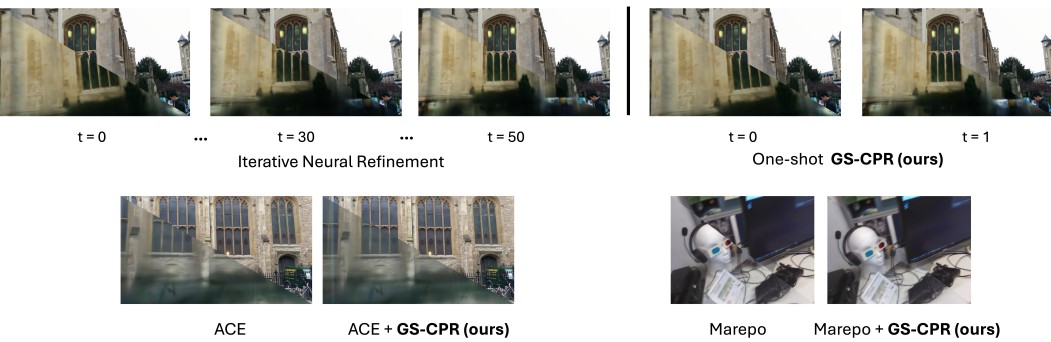

Figure 1: GS-CPR refines pose predictions of state-of-the-art APR and SCR models in a one-shot manner, achieving greater accuracy compared to the iterative neural refinement method, such as NeFeS (Chen et al., 2024a). Each subfigure is divided by a diagonal line, with the **bottom left** part rendered using the estimated/refined pose and the **top right** part displaying the ground truth image.

## Abstract

We leverage 3D Gaussian Splatting (3DGS) as a scene representation and propose a novel test-time camera pose refinement (CPR) framework, GS-CPR. This framework enhances the localization accuracy of state-of-the-art absolute pose regression and scene coordinate regression methods. The 3DGS model renders high-quality synthetic images and depth maps to facilitate the establishment of 2D-3D correspondences. GS-CPR obviates the need for training feature extractors or descriptors by operating directly on RGB images, utilizing the 3D foundation model, MASt3R, for precise 2D matching. To improve the robustness of our model in challenging outdoor environments, we incorporate an exposure-adaptive module within the 3DGS framework. Consequently, GS-CPR enables efficient one-shot pose refinement given a single RGB query and a coarse initial pose estimation. Our proposed approach surpasses leading NeRF-based optimization methods in both accuracy and runtime across indoor and outdoor visual localization benchmarks, achieving new state-of-the-art accuracy on two indoor datasets. The project page is available at: `https://xrim-lab.github.io/GS-CPR/`.

## 1 Introduction

Camera relocalization, the task of determining the 6-DoF camera pose within a given environment based on a query image, is critical for numerous applications, including robotics, autonomous vehicles, augmented reality, and virtual reality. Current methods for camera pose estimation primarily fall into the categories of structure-based approaches and absolute pose regression (APR) techniques. Classic structure-based pipelines (Dusmanu et al., 2019; Sarlin et al., 2019; Taira et al., 2018; Noh et al., 2017; Sattler et al., 2016; Sarlin et al., 2020; Lindenberger et al., 2023) rely on 2D-3D correspondences between a point cloud and the reference image. Another class of structure-based

---

[*]cliudg@connect.ust.hk, research conducted during a visit at Active Vision Lab, University of Oxford.

methods - Scene Coordinate Regression (SCR) (Brachmann et al., 2017; 2023; Wang et al., 2024a; Brachmann & Rother, 2021) - uses neural networks for direct regression of 2D-3D correspondences. These 2D-3D correspondences are fed into Perspective-n-Point (PnP) (Gao et al., 2003) for pose estimation. APR methods (Kendall et al., 2015; Wang et al., 2019; Chen et al., 2021; Shavit et al., 2021) employ neural networks to infer camera poses from query images directly. While APR approaches offer fast inference times, they often struggle with accuracy and generalization (Sattler et al., 2019; Liu et al., 2024a). SCR methods generally achieve higher accuracy but at the cost of increased computational complexity.

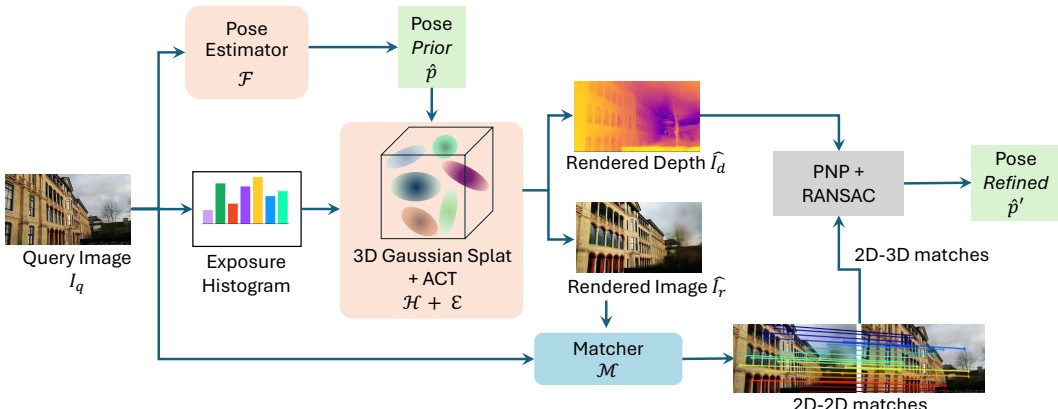

Figure 2: Overview of GS-CPR. We assume the availability of a pre-trained pose estimator $\mathcal{F}$ and a pre-trained 3DGS model $\mathcal{H}$ of the scene. For a query image $I_q$, we first obtain an initial estimated pose $\hat{p}$ from the pose estimator $\mathcal{F}$. Our goal is to output a refined pose $\hat{p}'$.

Given the above limitations, there has been a growing interest in pose refinement methods to enhance the accuracy of the *initial* pose estimates of an underlying pose-estimation method. Recent approaches have leveraged Neural Radiance Fields (NeRF) (Mildenhall et al., 2020) for this purpose. For instance, NeFeS (Chen et al., 2024a) proposes a test-time refinement pipeline. However, it offers limited improvements in accuracy and suffers from slow convergence due to the computational demands of NeRF rendering and the requirement for backpropagation through the pose estimation model. Furthermore, a recent NeRF-based localization method - CrossFire (Moreau et al., 2023) - establishes explicit 2D-3D matches using features rendered from NeRF. However, training a customized scene model together with the scene-specific localization descriptor is required, and it exhibits lower accuracy compared to classic structure-based methods.

To address the challenges of slow convergence, limited accuracy, and the need for training customized feature descriptors, we propose a novel test-time pose refinement framework, termed GS-CPR, as illustrated in Figure 1 and Figure 2. GS-CPR employs 3D Gaussian Splatting (3DGS) (Kerbl et al., 2023) for scene representation and leverages its high-quality, fast novel view synthesis (NVS) capabilities to render images and depth maps. This facilitates the efficient establishment of 2D-3D correspondences between the query image and the rendered image, based on the initial pose estimate from the underlying pose estimator (e.g., APR, SCR). We incorporate an exposure-adaptive module into the 3DGS model to improve its robustness to the domain shift between the query image and the rendered image. Secondly, our method operates directly on RGB images, utilizing the 3D vision foundation model MASt3R (Leroy et al., 2024) for precise matching, eliminating the need for training scene-specific feature extractors or descriptors (Chen et al., 2024a; Moreau et al., 2023). This significantly accelerates our method compared to iterative NeRF-based refinement methods (Chen et al., 2024a) and makes our framework easier to deploy than CrossFire (Moreau et al., 2023) and its variants (Zhou et al., 2024; Liu et al., 2023; Zhao et al., 2024).

Lastly, we conduct comprehensive quantitative evaluations and ablation studies on the 7Scenes (Glocker et al., 2013; Shotton et al., 2013), 12Scenes (Valentin et al., 2016), and Cambridge Landmarks (Kendall et al., 2015) benchmarks. GS-CPR significantly enhances the pose estimation accuracy of both APR and SCR methods across these benchmarks, achieving new state-of-the-art accuracy on the two indoor datasets. Unlike previous NeRF-based methods (Chen et al., 2024a),

which fail to improve SCR methods, such as ACE (Brachmann et al., 2023), our method offers substantial improvements and outperforms other leading NeRF-based methods (Germain et al., 2022; Moreau et al., 2023; Zhou et al., 2024; Liu et al., 2023; Zhao et al., 2024).

## 2    RELATED WORK

**Pose Estimation without 3D Representation.** A straightforward approach for coarse pose estimation is using image retrieval (Arandjelovic et al., 2016; Ge et al., 2020; Gordo et al., 2017) to average poses from top-retrieved images, but this lacks precision. Absolute Pose Regression (APR) methods (Kendall et al., 2015; Kendall & Cipolla, 2016; 2017; Wang et al., 2019; Chen et al., 2021; 2022; Shavit et al., 2021; Chen et al., 2024b; Lin et al., 2024) directly regress a pose from a query image using trained models, bypassing 3D representations and geometric relationships. Despite being fast, APR methods suffer in accuracy and generalization (Sattler et al., 2019; Liu et al., 2024a) compared to structure-based techniques. LENS (Moreau et al., 2022) enhances APR by augmenting views with NeRF, but matching the accuracy of 3D structure-based methods remains challenging. To improve APR methods' accuracy, we used 3DGS as a 3D representation and utilized its geometry information to optimize the initial prediction.

**Structure-based Pose Estimation.** Classical 3D structure-based methods, like the hierarchical localization pipeline (HLoc) (Dusmanu et al., 2019; Sarlin et al., 2019; Taira et al., 2018; Noh et al., 2017; Sattler et al., 2016; Sarlin et al., 2020; Lindenberger et al., 2023), predict camera poses using a point cloud and a database of reference images, requiring descriptor storage and 2D-3D correspondence through image retrieval. In contrast, Scene Coordinate Regression (SCR) methods (Brachmann et al., 2017; 2023; Wang et al., 2024a; Brachmann & Rother, 2021) directly regress 2D-3D correspondences using neural networks and apply PnP (Gao et al., 2003) and RANSAC (Fischler & Bolles, 1981) for pose estimation. Our GS-CPR eliminates the need for reference images and descriptor databases by using a 3DGS model for scene representation, further optimizing SCR outputs like ACE (Brachmann et al., 2023).

**NeRF-based Pose Estimation.** NeRF-based pose estimation methods (Chen et al., 2024a; Yen-Chen et al., 2021; Lin et al., 2023) rely on iterative rendering and pose updates, leading to slow convergence and limited accuracy. While NeFeS (Chen et al., 2024a) improves APR pose estimation, it faces difficulties in enhancing SCR results and suffers from long refinement runtime. HR-APR (Liu et al., 2024a) speeds up optimization by 30%, but the average runtime of each query still takes several seconds on a high-performance GPU. Other NeRF-based methods like FQN (Germain et al., 2022), CrossFire (Moreau et al., 2023), NeRFLoc (Liu et al., 2023), and NeRFMatch (Zhou et al., 2024) improve positioning by establishing 2D-3D matches but require specialized feature extractors and suffer from slow rendering and quality issues.

**3DGS-based Pose Estimation.** With the NVS field transitioning from NeRF to 3DGS, methods proposed by  Sun et al. (2023) and  Botashev et al. (2024) refine camera poses in an inefficient iterative manner by inverting 3DGS, following iNeRF (Yen-Chen et al., 2021). In contrast, 6DGS (Bortolon et al., 2024) achieves a one-shot estimate by projecting rays from an ellipsoid surface, avoiding iteration. Although both methods use 3DGS for visual localization, neither has been tested on large benchmarks (Kendall et al., 2015; Valentin et al., 2016) or compared with mainstream methods like SCR and APR. We propose an approach using 3DGS for 2D-3D correspondences, similar to CrossFire (Moreau et al., 2023), but without requiring training feature extractors or feature matchers. Our method generates high-quality synthetic images and employs direct 2D-2D matching, making it faster and easier to deploy than previous NeRF-based methods such as NeFeS, CrossFire, and other variants (Germain et al., 2022; Zhou et al., 2024; Liu et al., 2023; 2024a; Zhao et al., 2024).

## 3    PROPOSED METHOD

GS-CPR is a test-time camera pose refinement framework. We assume the availability of a pre-trained pose estimator and a 3DGS model of the scene. For a query image, we first obtain an initial estimated pose from the pose estimator. Our goal is to output a refined pose.

Given a query image $I_q \in \mathbb{R}^{H \times W \times 3}$ with camera intrinsics $K \in \mathbb{R}^{3 \times 3}$, a pose estimator $\mathcal{F}$ (typically an APR or SCR model) predicts an *initial* 6-DoF pose $\hat{p} = [\hat{\mathbf{t}} | \hat{\mathbf{R}}]$, where $\hat{\mathbf{t}} \in \mathbb{R}^3$ and $\hat{\mathbf{R}} \in \mathbb{R}^{3 \times 3}$

represent the estimated translation and rotation respectively. Subsequently, for the viewpoint $\hat{p}$, a pretrained 3DGS model $\mathcal{H}$ renders an image $\hat{I}_r \in \mathbb{R}^{H \times W \times 3}$ and a depth map $\hat{I}_d \in \mathbb{R}^{H \times W \times 1}$. We use an exposure-adaptive affine color transformation (ACT) module $\mathcal{E}$ during this rendering process to enhance the robustness of our model to challenging outdoor environments (see Section 3.1). A matcher $\mathcal{M}$ then establishes dense 2D-2D correspondences between $I_q$ and $\hat{I}_r$. Then we can establish the 2D-3D matches based on $\hat{I}_q$ and $\hat{I}_d$ (see Section 3.2). Finally, we obtain the refined pose $\hat{p}'$ from these 2D-3D matches (see Section 3.2). An overview of our framework is depicted in Figure 2. We also explore a faster pose refinement framework without 2D-3D matches depicted in Figure 3 (see Section 3.3).

### 3.1 3DGS Test-time Exposure Adaptation

Existing literature (Kerbl et al., 2023; Lu et al., 2024) shows that 3DGS achieves high-quality novel view renderings but assumes training and testing without significant photometric distortions. In visual relocalization, mapping and query sequences often differ in lighting due to varying times, weather, and exposure. This creates a significant appearance gap between 3DGS renderings and query images, negatively impacting 2D-2D matching performance.

To address this issue, we apply an exposure-adaptive affine color transformation module $\mathcal{E}$ (Chen et al., 2022; 2024a) to 3DGS, allowing the 3DGS to adaptively render appearances during testing and accurately reflect the exposure of $I_q$. Specifically, we use a 4-layer MLP that takes the luminance histogram of the query image as input and produces a 3x3 matrix $\mathbf{Q}$ along with a 3-dimensional bias vector $\mathbf{b}$. These outputs are then directly applied to the rendered pixels of the 3DGS as shown in Equation 1, ensuring a closer match to the exposure of the query image.

$$\hat{\mathbf{C}}(\mathbf{r}) = \mathbf{Q}\hat{\mathbf{C}}_{\text{rend}}(\mathbf{r}) + \mathbf{b}, \tag{1}$$

where $\hat{\mathbf{C}}(\mathbf{r})$ is the final per-pixel color and $\hat{\mathbf{C}}_{\text{rend}}(\mathbf{r})$ is the rendered per-pixel color obtained from the 3DGS model $\mathcal{H}$.

### 3.2 Pose Refinement with 2D-3D Correspondences

GS-CPR estimates the camera pose by establishing 2D-3D correspondences between the query image $I_q$ and the scene representation. This process involves the following steps:

**2D-2D Matching.** First, an image $\hat{I}_r$ is rendered from the initial estimated viewpoint $\hat{p}$. A Matcher $\mathcal{M}$ is then used to establish 2D-2D pixel correspondences $C_{q,r}$ between the query image $I_q$ and the rendered image $\hat{I}_r$. In our implementation, the matcher $\mathcal{M}$ is a recently released 3D vision foundation model, MASt3R (Leroy et al., 2024). MASt3R demonstrates strong robustness for 2D-2D matching across image pairs with the sim-to-real domain gap.

**3D Coordinate Map Generation.** Simultaneously, we use our trained 3DGS model $\mathcal{H}$ to render a depth map $\hat{I}_d$ from the viewpoint $\hat{p}$. We modify the rasterization engine of 3DGS to render the depth map as follows:

$$\hat{I}_d = \sum_{i \in N} d_i \alpha_i \prod_{j=1}^{i-1} (1 - \alpha_j), \tag{2}$$

where $d_i$ is the z-depth of each Gaussian in the viewspace and $\alpha_i$ is the learned opacity multiplied by the projected 2D covariance of the $i^{th}$ Gaussian. In our framework, ground truth depth maps are not required for supervision during training of the 3DGS model $\mathcal{H}$. Using the rendered depth map $\hat{I}_d$, camera intrinsics $K$, and pose $\hat{p}$, we obtain the 3D coordinate map $X_r^d \in \mathbb{R}^{H \times W \times 3}$ for the rendered image $\hat{I}_r$.

**Establishing 2D-3D Correspondences.** By combining the 2D-2D correspondences $C_{q,r}$ with the 3D coordinate map $X_r^d$, we establish 2D-3D correspondences between $I_q$ and the scene. For each matched pixel in $I_q$, we obtain its corresponding 3D coordinate from $X_r^d$.

**Pose Refinement.** Finally, we obtain the refined pose $\hat{p}'$ by feeding these 2D-3D correspondences into a PnP (Gao et al., 2003) solver with RANSAC (Fischler & Bolles, 1981) loop. This process

does not require backpropagation through the pose estimator $\mathcal{F}$ or the 3DGS model $\mathcal{H}$, ensuring efficient computation and enabling its usage with any black-box pose estimator model.

Using 2D-3D correspondences, coupled with PnP + RANSAC, provides a robust pose refinement that is much faster and more accurate than methods relying solely on rendering and comparison (Yen-Chen et al., 2021; Lin et al., 2023; Sun et al., 2023). Furthermore, our method eliminates the requirement of training specialized feature descriptors that previous approaches (Chen et al., 2024a; Moreau et al., 2023; Chen et al., 2022; Zhao et al., 2024) rely on for robustness.

### 3.3 Faster Alternative with Relative Post Estimation

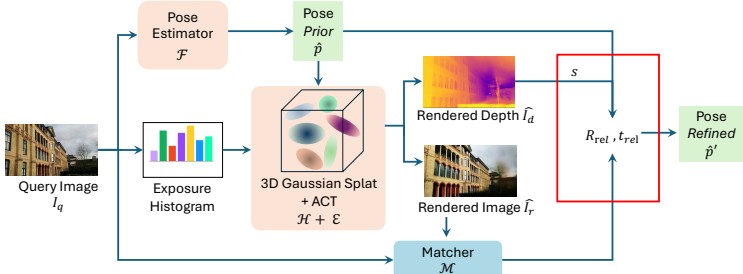

Figure 3: Overview of GS-CPR$_{\text{rel}}$. Different from GS-CPR in Figure 2 (highlight with the red box), we use $\hat{I}_d$ to recover the scale $s$ of $\mathbf{t}_{\text{rel}}$. Then we calculate the refined pose $\hat{p}'$ based on $\mathbf{R}_{\text{rel}}$ and $s\mathbf{t}_{\text{rel}}$ without matching.

While GS-CPR provides high accuracy through 2D-3D correspondences, we also explore an alternative approach that prioritizes computational efficiency. This variant, which we call GS-CPR$_{\text{rel}}$, utilizes MASt3R's point map registration capabilities to estimate relative pose without matching. Figure 3 shows an overview of the GS-CPR$_{\text{rel}}$ approach.

Specifically, MASt3R generates point maps $\mathbf{P}_q$ and $\mathbf{P}_r$ for both the query image $I_q$ and the rendered image $\hat{I}_r$ and predicts the relative rotation $\mathbf{R}_{\text{rel}}$ and translation $\mathbf{t}_{\text{rel}}$ between the two images. However, this relative pose predicted by MASt3R needs to be aligned to the scene's scale $s$. We recover the scale by aligning the point map $\mathbf{P}_r$ with the depth map $\hat{I}_d$ rendered from the 3DGS model $\mathcal{H}$. The final refined pose $\hat{p}'$ is computed as:

$$\hat{p}' = [\hat{\mathbf{R}}'|\hat{\mathbf{t}}'] = [\mathbf{R}_{\text{rel}}\hat{\mathbf{R}}|\mathbf{R}_{\text{rel}}\hat{\mathbf{t}} + s\mathbf{t}_{\text{rel}}], \tag{3}$$

where $\hat{\mathbf{R}}$, $\hat{\mathbf{t}}$ are the initial rotation and translation estimates. As shown in Table 5 and 6, GS-CPR$_{\text{rel}}$ offers a trade-off between speed and accuracy, making it ideal for rapid refinement of APR methods like DFNet (Chen et al., 2022).

## 4 Experiments

### 4.1 Evaluation Setup

**Datasets.** We evaluate the performance of GS-CPR across three widely used public visual localization datasets. The 7Scenes dataset (Glocker et al., 2013; Shotton et al., 2013) comprises seven indoor scenes with volumes ranging from $1$–$18\,\text{m}^3$. The 12Scenes dataset (Valentin et al., 2016) features 12 larger indoor scenes, with volumes spanning from $14$–$79\,\text{m}^3$. The Cambridge Landmarks dataset (Kendall et al., 2015) represents large-scale outdoor scenarios, characterized by challenges such as moving objects and varying lighting conditions between query and training images.

**Evaluation Metrics.** We report two types of metrics to compare the performance of different methods. The first metric is the median translation and rotation error. The second metric is the recall rate, which measures the percentage of test images localized within $a$ cm and $b°$.

**Baselines.** In our experiment, to demonstrate the improvement capabilities of our framework, we use the initial estimates of APR and SCR methods as our baseline. We employ our method on top of

Table 1: Comparisons on 7Scenes dataset. The median translation and rotation errors (cm/°) of different methods. The best results are in **bold** (lower is better). Second best results are indicated with an underline. NRP stands for Neural Rendering-based Pose Estimation.

|  | Methods | Chess | Fire | Heads | Office | Pumpkin | Redkitchen | Stairs | Avg. ↓ [cm/°] |
|---|---|---|---|---|---|---|---|---|---|
| APR | PoseNet (Kendall et al., 2015) | 10/4.02 | 27/10.0 | 18/13.0 | 17/5.97 | 19/4.67 | 22/5.91 | 35/10.5 | 21/7.74 |
|  | MS-Transformer (Shavit et al., 2021) | 11/6.38 | 23/11.5 | 13/13.0 | 18/8.14 | 17/8.42 | 16/8.92 | 29/10.3 | 18/9.51 |
|  | DFNet (Chen et al., 2022) | 3/1.12 | 6/2.30 | 4/2.29 | 6/1.54 | 7/1.92 | 7/1.74 | 12/2.63 | 6/1.93 |
|  | Marepo (Chen et al., 2024b) | 1.9/0.83 | 2.3/0.92 | 2.1/1.24 | 2.9/0.93 | 2.5/0.88 | 2.9/0.98 | 5.9/1.48 | 2.9/1.04 |
| SCR | DSAC* (Brachmann & Rother, 2021) | **0.5**/0.17 | 0.8/0.28 | 0.5/0.34 | 1.2/0.34 | 1.2/0.28 | **0.7**/0.21 | 2.7/0.78 | 1.1/0.34 |
|  | ACE (Brachmann et al., 2023) | **0.5**/0.18 | 0.8/0.33 | 0.5/0.33 | 1.0/0.29 | 1.0/**0.22** | 0.8/0.2 | 2.9/0.81 | 1.1/0.34 |
|  | GLACE (Wang et al., 2024a) | 0.6/0.18 | 0.9/0.34 | 0.6/0.34 | 1.1/0.29 | **0.9**/0.23 | 0.8/0.20 | 3.2/0.93 | 1.2/0.36 |
| NRP | FQN-MN (Germain et al., 2022) | 4.1/1.31 | 10.5/2.97 | 9.2/2.45 | 3.6/2.36 | 4.6/1.76 | 16.1/4.42 | 139.5/34.67 | 28/7.3 |
|  | CrossFire (Moreau et al., 2023) | 1/0.4 | 5/1.9 | 3/2.3 | 5/1.6 | 3/0.8 | 2/0.8 | 12/1.9 | 4.4/1.38 |
|  | pNeRFLoc (Zhao et al., 2024) | 2/0.8 | 2/0.88 | 1/0.83 | 3/1.05 | 6/1.51 | 5/1.54 | 32/5.73 | 7.3/1.76 |
|  | DFNet + NeFeS$_{50}$ (Chen et al., 2024a) | 2/0.57 | 2/0.74 | 2/1.28 | 2/0.56 | 2/0.55 | 2/0.57 | 5/1.28 | 2.4/0.79 |
|  | HR-APR (Liu et al., 2024a) | 2/0.55 | 2/0.75 | 2/1.45 | 2/0.64 | 2/0.62 | 2/0.67 | 5/1.30 | 2.4/0.85 |
|  | NeRFMatch (Zhou et al., 2024) | 0.9/0.3 | 1.1/0.4 | 1.5/1.0 | 3.0/0.8 | 2.2/0.6 | 1.0/0.3 | 10.1/1.7 | 2.8/0.7 |
|  | MCLoc (Trivigno et al., 2024) | 2/0.8 | 3/1.4 | 3/1.3 | 4/1.3 | 5/1.6 | 6/1.6 | 6/2.0 | 4.1/1.43 |
|  | DFNet + **GS-CPR (ours)** | 0.7/0.20 | 0.9/0.32 | 0.6/0.36 | 1.2/0.32 | 1.3/0.31 | 0.9/0.25 | 2.2/0.61 | 1.1/0.34 |
|  | Marepo + **GS-CPR (ours)** | 0.6/0.18 | 0.7/0.28 | 0.5/0.32 | 1.1/0.29 | 1.0/0.26 | 0.8/0.21 | 1.5/0.44 | 0.9/0.28 |
|  | ACE + **GS-CPR (ours)** | **0.5**/**0.15** | **0.6**/**0.25** | **0.4**/**0.28** | **0.9**/**0.26** | 1.0/0.23 | 0.7/**0.17** | **1.4**/**0.42** | **0.8**/**0.25** |

the prevailing APR methods, DFNet (Chen et al., 2022) and Marepo (Chen et al., 2024b), as well as a well-known SCR method, ACE (Brachmann et al., 2023), as the pose estimator $\mathcal{F}$. We follow the default settings of these pose estimators to obtain the initial pose prior for each query image[1]. The term *APR/SCR + GS-CPR* denotes the one-shot refinement. A similar naming convention applies to *APR/SCR + GS-CPR$_{rel}$*. We also include a comparison here with the state-of-the-art NeRF-based methods (Chen et al., 2024a; Moreau et al., 2023; Zhou et al., 2024; Liu et al., 2024a; Germain et al., 2022; Zhao et al., 2024; Liu et al., 2023) and MCLoc (Trivigno et al., 2024), which is a pose refinement framework agnostic to scene representation. MCLoc provides results using 3DGS models as scene representations for the 7Scenes and Cambridge datasets.

**Implementation Details.** GT Poses: For both the 7Scenes and 12Scenes datasets, we adopt the SfM ground truth (GT) provided by Brachmann et al. (2021). As demonstrated in NeFeS (Chen et al., 2024a), SfM GT can render superior geometric details compared to dSLAM GT for the 7Scenes dataset. Gaussian Splatting: For the training of the 3DGS model of each scene, we utilize the sparse point cloud of training frames generated by COLMAP (Schonberger & Frahm, 2016) as the initial input. We select Scaffold-GS (Lu et al., 2024) as our 3DGS representation, incorporating modifications detailed in Sections 3.1 and 3.2 to adapt exposure and enable depth rendering. Scaffold-GS reduces redundant Gaussians while delivering high-quality rendering compared to the vanilla 3DGS (Kerbl et al., 2023). For the exposure-adaptive ACT module, we follow the default setting in Chen et al. (2024a), computing the query image's histogram in the YUV color space and binning the luminance channel into 10 bins. In addition, we apply temporal object filtering to filter out moving objects in the dynamic scene using an off-the-shelf method (Cheng et al., 2022), leading to better accuracy in scene reconstruction quality and pixel-matching performance. Training Details: We employ the official pre-trained MASt3R (Leroy et al., 2024) model without fine-tuning for 2D-2D matching and resize all images to 512 pixels on their largest dimension. The modified Scaffold-GS model is trained for each scene with 30,000 iterations on an NVIDIA A6000 GPU. We implement our framework with PyTorch (Paszke et al., 2019). Additional details can be found in the Appendix A.1 and A.2.

## 4.2 Localization Accuracy

We conduct quantitative experiments on three datasets to evaluate the improved localization accuracy of our framework compared to the APR and SCR methods.

**7Scenes Dataset.** Using the 7Scenes dataset, we evaluate the performance of DFNet, Marepo, and ACE with GS-CPR. Table 1 demonstrates that GS-CPR significantly reduces pose estimation errors for DFNet, Marepo, and ACE with one-shot refinement. Table 2 shows that GS-CPR significantly improves the proportion of query images below 5cm, 5° and 2cm, 2° pose error. It is worth noting

---

[1]Note that the original paper of Marepo reports results on 7Scenes using dSLAM GT; we retrained the ACE head of Marepo using SfM GT.

Table 2: We report the average percentage (%) of frames below a $(5cm, 5°)$ and $(2cm, 2°)$ pose error across 7Scenes. IR denotes image retrieval.

|  | Methods | Avg. ↑ [5cm, 5°] | Avg. ↑ [2cm, 2°] |
|---|---|---|---|
| APR | DFNet | 43.1 | 8.4 |
|  | Marepo | 84.0 | 33.7 |
| IR+SfM points | HLoc (SP + SG) (Sarlin et al., 2020; 2019) | 95.7 | 84.5 |
|  | DVLAD+R2D2 (Torii et al., 2015; Revaud et al., 2019) | 95.7 | 87.2 |
| SCR | DSAC* | 97.8 | 80.7 |
|  | ACE | 97.1 | 83.3 |
|  | GLACE | 95.6 | 82.2 |
| NRP | DFNet + NeFeS$_{50}$ | 78.3 | 45.9 |
|  | HR-APR | 76.4 | 40.2 |
|  | NeRFMatch | 78.4 | - |
|  | NeRFLoc (Liu et al., 2023) | 89.5 | - |
|  | DFNet + **GS-CPR (ours)** | 94.2 | 76.5 |
|  | Marepo + **GS-CPR (ours)** | 99.4 | 89.6 |
|  | ACE + **GS-CPR (ours)** | **100** | **93.1** |

Table 3: Comparisons on Cambridge Landmarks dataset. We report the median translation and rotation errors (cm/°) of different methods. Best results are in **bold** (lower is better) among the NRP approaches.

|  | Methods | Kings | Hospital | Shop | Church | Avg. ↓ [cm/°] |
|---|---|---|---|---|---|---|
| IR + SfM points | HLoc (SP+SG) (k=1) | 13/0.22 | 18/0.38 | 6/0.25 | 9/0.28 | 12/0.28 |
|  | HLoc (SP+SG) (k=10) | 11/0.2 | 15/0.31 | 4/0.21 | 7/0.22 | 9/0.24 |
| APR | PoseNet | 93/2.73 | 224/7.88 | 147/6.62 | 237/5.94 | 175/5.79 |
|  | MS-Transformer | 85/1.45 | 175/2.43 | 88/3.20 | 166/4.12 | 129/2.80 |
|  | LENS (Moreau et al., 2022) | 33/0.5 | 44/0.9 | 27/1.6 | 53/1.6 | 39/1.15 |
|  | DFNet | 73/2.37 | 200/2.98 | 67/2.21 | 137/4.02 | 119/2.90 |
|  | PMNet (Lin et al., 2024) | 68/1.97 | 103/1.31 | 58/2.10 | 133/3.73 | 90/2.27 |
| SCR | ACE | 29/0.38 | 31/0.61 | 5/0.3 | 19/0.6 | 21/0.47 |
|  | GLACE[1] | 20/0.32 | 20/0.41 | 5/0.22 | 9/0.3 | 14/0.32 |
| NRP | FQN-MN | 28/0.4 | 54/0.8 | 13/0.6 | 58/2 | 38/1 |
|  | CrossFire | 47/0.7 | 43/0.7 | 20/1.2 | 39/1.4 | 37/1 |
|  | DFNet + NeFeS$_{30}$[2] | 37/0.64 | 98/1.61 | 17/0.60 | 42/1.38 | 49/1.06 |
|  | DFNet + NeFeS$_{50}$ | 37/0.54 | 52/0.88 | 15/0.53 | 37/1.14 | 35/0.77 |
|  | HR-APR | 36/0.58 | 53/0.89 | 13/0.51 | 38/1.16 | 35/0.78 |
|  | MCLoc | 31/0.42 | 39/0.73 | 12/0.45 | 26/0.8 | 27/0.6 |
|  | DFNet + **GS-CPR (ours)** | 23/0.32 | 42/0.74 | 10/0.36 | 27/0.62 | 26/0.51 |
|  | ACE + **GS-CPR (ours)** | **20/0.29** | **21/0.40** | **5/0.24** | **13/0.40** | **15/0.33** |

[1] We report the accuracy based on official open-source models (Wang et al., 2024a).
[2] Results of DFNet + NeFeS$_{30}$ taken from Liu et al. (2024a).

that ACE + GS-CPR outperforms HLoc (Superpoint (DeTone et al., 2018) + Superglue (Sarlin et al., 2020)), indicating that 3DGS has the potential to replace traditional point-clouds in visual localization pipelines. Figure 4 (a) shows that after refinement using our GS-CPR, the rendered image of the estimated pose better matches the real image.

**Cambridge Landmarks Dataset.** We conduct a quantitative evaluation by deploying DFNet and ACE with GS-CPR. Marepo is not included in this comparison due to the absence of an official model for this dataset. Table 3 demonstrates that GS-CPR significantly reduces pose estimation errors for both DFNet and ACE. Specifically, the accuracy of DFNet + GS-CPR with one-shot optimization significantly surpasses that of CrossFire and DFNet + NeFeS with 30 and even 50 steps of optimization (see Table 3). This result fully demonstrates the efficiency of our GS-CPR. On the Kings College scene, DFNet + GS-CPR outperforms ACE after our refinement. ACE + GS-CPR consistently improves ACE accuracy across all four scenes. Refining the pose using our method results in a rendered image that aligns more accurately with the ground truth image as illustrated in Figure 4 (c).

**12Scenes Dataset.** We conduct the quantitative evaluation using Marepo and ACE with GS-CPR. The former works (Brachmann et al., 2023; Wang et al., 2024a) report the percentage of frames below a $5cm, 5°$ pose error. Since SCR methods have already achieved good results with this metric, in this paper we use a more stringent standard $(2cm, 2°)$ and report the median translation and

Table 4: We report the average accuracy (%) of frames meeting a $[5\text{cm}, 5°]$, $[2\text{cm}, 2°]$ pose error threshold, and the median translation and rotation errors (cm/°) across 12Scenes.

| Methods | Avg. Err ↓ [cm/°] | Avg. ↑ [5cm, 5°] | Avg. ↑ [2cm, 2°] |
|---|---|---|---|
| Marepo | 2.1/1.04 | 95 | 50.4 |
| DSAC* | **0.5**/0.25 | 99.8 | 96.7 |
| ACE | 0.7/0.26 | **100** | 97.2 |
| GLACE | 0.7/0.25 | **100** | 97.5 |
| Marepo + **GS-CPR (ours)** | 0.7/0.28 | 98.9 | 90.9 |
| ACE + **GS-CPR (ours)** | **0.5/0.21** | **100** | **98.7** |

Table 5: We report the average accuracy (%) of frames meeting a $[5\text{cm}, 5°]$ pose error threshold, and the median translation and rotation errors (cm/°).

| Datasets | 7Scenes | | Cambridge |
|---|---|---|---|
| Methods | Avg. Acc ↑ [5cm, 5°] | Avg. Err ↓ [cm/°] | Avg. Err ↓ [cm/°] |
| DFNet | 43.1 | 6/1.93 | 119/2.9 |
| DFNet + **GS-CPR$_{rel}$ (ours)** | 80.5 | 2.7/0.38 | 55/0.57 |
| DFNet + **GS-CPR (ours)** | **94.2** | **1.1/0.34** | **26/0.51** |
| ACE | 97.1 | 1.1/0.34 | 21/0.47 |
| ACE + **GS-CPR$_{rel}$ (ours)** | 79.9 | 2.8/0.43 | 47/0.54 |
| ACE + **GS-CPR (ours)** | **100** | **0.8/0.25** | **15/0.33** |

rotation errors (cm/°). Table 4 shows that GS-CPR significantly improves the percentage of query images below $2\text{cm}, 2°$ pose error and median pose error for Marepo and ACE. Figure 4 (b) shows that after refinement using our GS-CPR, the rendered image with our pose estimation aligns better with the real image.

**GS-CPR vs. GS-CPR$_{rel}$.** We compare GS-CPR, a pose refinement framework that uses 2D-3D correspondence, with GS-CPR$_{rel}$, a faster alternative that uses relative pose from MASt3R. Both frameworks are evaluated on 7Scenes and Cambridge Landmarks datasets using DFNet and ACE predictions. Table 5 shows that GS-CPR$_{rel}$ achieves notable accuracy improvement with DFNet on both indoor and outdoor datasets, though it is less effective than GS-CPR. However, GS-CPR$_{rel}$ is significantly faster than GS-CPR and other NeRF-based methods, as discussed in Section 4.3. While GS-CPR$_{rel}$ improves coarse pose estimates from APR methods like DFNet, it struggles with accurate pose estimates from SCR methods. For ACE, GS-CPR$_{rel}$ results in performance degradation because our pose refinement relies on the relative pose estimator MASt3R, which struggles to provide more accurate relative pose estimates when the ACE-predicted pose is sufficiently close to the GT pose. Higher median rotation and translation errors in Table 5 compared to GS-CPR indicate that scale recovery is not the only challenge for GS-CPR$_{rel}$, as rotation is scale-independent.

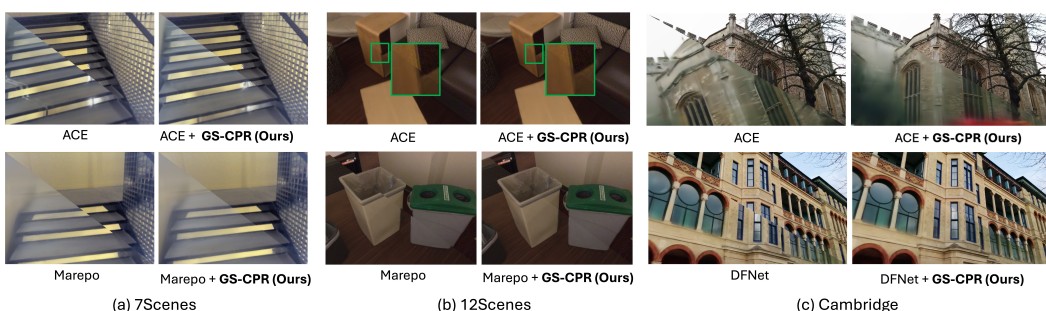

Figure 4: Our GS-CPR enhances pose predictions for Marepo, DFNet, and ACE. Each subfigure is divided by a diagonal line, with the **bottom left** part rendered using the estimated/refined pose and the **top right** part displaying the ground truth image. Patches highlighting visual differences are emphasized with green insets for enhanced visibility.

Table 6: Runtime Analysis (test on Cambridge Landmarks).

| Methods | CrossFire | DFNet + NeFeS$_{50}$ | HR-APR | MCLoc | DFNet + GS-CPR$_{rel}$ (ours) | DFNet + GS-CPR (ours) | ACE + GS-CPR (ours) |
|---|---|---|---|---|---|---|---|
| Avg. ↓ [cm/°] | 37/1.0 | 35/0.8 | 35/0.8 | 27/0.6 | 55/0.6 | 26/0.5 | **15/0.3** |
| Avg. ↓ time (s) | 0.3 | 10 | 8.5 | 2.4 | **0.08** | 0.18 | 0.19 |

Table 7: Results of different matchers (LoFTR (Sun et al., 2021), DUSt3R (Wang et al., 2024b), and MASt3R (Leroy et al., 2024)) on the 7Scenes dataset. GS-CPR [L] denotes using LoFTR as the matcher $\mathcal{M}$, GS-CPR [D] denotes using DUSt3R as $\mathcal{M}$, and GS-CPR [M] denotes using MASt3R as $\mathcal{M}$. The table presents median translation and rotation errors (cm/°) of the different methods.

| Methods | Marepo | + GS-CPR [L] | + GS-CPR [D] | + GS-CPR [M] | ACE | + GS-CPR [L] | + GS-CPR [D] | + GS-CPR [M] |
|---|---|---|---|---|---|---|---|---|
| Avg. ↓ [cm/°] | 2.9/1.04 | 1.5/0.40 | 2.1/0.7 | **0.9/0.28** | 1.1/0.34 | 1.0/0.31 | 1.5/0.6 | **0.8/0.25** |

## 4.3 RUNTIME ANALYSIS

We evaluate the runtime of the proposed framework using an NVIDIA GeForce RTX 4090 GPU. On average, 3DGS rendering takes 3.7 ms on the 7Scenes dataset and 12 ms on the Cambridge Landmarks dataset (due to higher scene complexity and image resolution). MASt3R relative pose estimation takes 71 ms. MASt3R matching takes an additional 42 ms, and PnP+RANSAC takes another 52 ms. As a result, our GS-CPR$_{rel}$ only adds 71 ms to the inference time of the pose estimator $\mathcal{F}$, and our GS-CPR adds less than 180 ms overhead. All time measurements are averaged over 1,000 runs. We compare the runtime and accuracy with other methods in Table 6. On the Cambridge Landmarks dataset, MCLoc requires an average of 2.4 s per query with 80 iterations (Trivigno et al., 2024). In contrast, our ACE+GS-CPR with one-shot optimization only takes 0.19 s per query. Therefore, in terms of efficiency and improvement, our GS-CPR is better than MCLoc when using 3DGS as scene representation. Although GS-CPR$_{rel}$ is less accurate than GS-CPR, it is more efficient. GS-CPR$_{rel}$ provides a feasible solution to pose refinement when the time budget is important.

## 4.4 ABLATION STUDY

In this section, we first demonstrate the rationale behind selecting MASt3R as the matcher $\mathcal{M}$ in GS-CPR. Subsequently, we show that ACT effectively reduces the domain gap between the query image and the rendered image, thereby enhancing the refinement accuracy.

**Different Matchers.** We compare three matching methods: LoFTR (Sun et al., 2021), DUSt3R (Wang et al., 2024b), and MASt3R – within GS-CPR in the 7Scenes dataset. For DUSt3R and MASt3R, we resize all images to 512 pixels in their largest dimension. For LoFTR, we use the pre-trained model for indoor scenes and maintain the frames in the 7Scenes dataset at $640 \times 480$. As shown in Table 7, Marepo + GS-CPR and ACE + GS-CPR using MASt3R as $\mathcal{M}$ achieve the highest improvement. Conversely, ACE + GS-CPR using DUSt3R does not yield any improvement. Marepo + GS-CPR using DUSt3R and Marepo/ACE + GS-CPR using LoFTR show a lower improvement compared to MASt3R. These results validate our choice of design to use MASt3R as a matcher $\mathcal{M}$.

**Affine Color Transformation.** To enhance the robustness of the 3DGS model in image rendering and to reduce the domain gap between the rendered image and the query image, we incorporated an ACT module into the Scaffold-GS model, as described in Section 3.1. Figure 5 illustrates the improvement in image rendering quality with the ACT module applied. The performance enhancement of GS-CPR from the ACT module is demonstrated in Table 8. On the Cambridge Landmarks dataset, employing the ACT module in the DFNet + GS-CPR setup reduces both the average median translation and rotation errors.

## 4.5 DISCUSSION

In this section, we provide additional insights and discussions of our design choices.

**Replace Feature Descriptors.** Given that 3DGS can render high-quality synthetic images $\hat{I}_r$ in real-time, we show that using a pre-trained 3D foundation model, MASt3R, can directly establish accurate 2D-2D correspondences $C_{q,r}$ between $I_q$ and $\hat{I}_r$ with a sim-to-real domain gap. As demon-

Table 8: Ablation study for ACT module on Cambridge Landmarks dataset. We report the median translation and rotation errors (cm/°).

| Methods | Kings | Hospital | Shop | Church | Avg. ↓ [cm/°] |
|---|---|---|---|---|---|
| DFNet + GS-CPR (w/o. ACT) | 34/0.46 | 54/0.84 | 12/**0.34** | 34/0.72 | 34/0.59 |
| DFNet + GS-CPR (w. ACT) | **23/0.32** | **42/0.74** | **10**/0.36 | **27/0.62** | **26/0.51** |

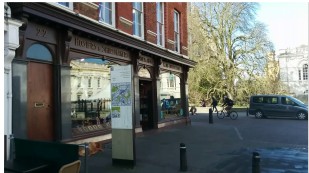 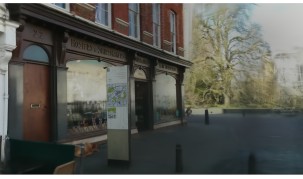 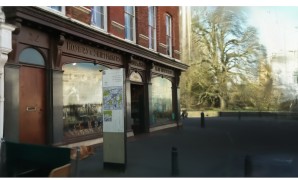

(a) Ground Truth      (b) Scaffold-GS/PSNR:16.5 dB      (c) Ours/PSNR:18.4 dB

Figure 5: Benefit of the ACT module. A regular 3DGS model tends to render images based on the lighting conditions and the appearance of its training frames, as demonstrated by the synthetic view of Scaffold-GS in (b). However, in challenging visual localization datasets, such as ShopFacade in the Cambridge Landmarks, some query frames may have different exposures compared to the training frames. (c) Our proposed Scaffold-GS + ACT can adaptively adjust the exposure based on the query's histogram.

strated in Section 4.2, GS-CPR achieves significantly higher accuracy than NeRF-based refinement pipelines that rely on feature rendering. Direct RGB matching makes our framework more compact, reduces runtime, eliminates the need for training additional neural radiance features, and simplifies both deployment and usage.

**Efficient and Effective Pose Refinement.** As a pose estimator, DFNet provides less accurate predictions than Marepo and ACE, but NeFeS reports the best results over DFNet. To ensure a fair comparison with NeFeS, we present examples in Figure 6 illustrating that our GS-CPR outperforms NeFeS in both efficiency and effectiveness. With only one-shot optimization, our GS-CPR achieves higher accuracy than NeFeS with 50 optimization iterations when combined with DFNet on both the indoor 7Scenes and outdoor Cambridge Landmarks datasets. This superior performance is due to our method's leverage of 3D geometry (depth rendering) of the representation, unlike previous NeRF-based refinement methods (Chen et al., 2024a; Yen-Chen et al., 2021) that use only 2D feature/photometric information in an iterative process, rendering candidate poses and comparing them with the target image. Additional discussion can be found in the Appendix A.3.

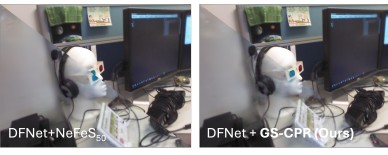 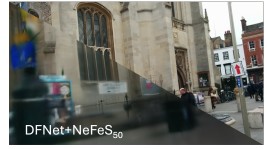 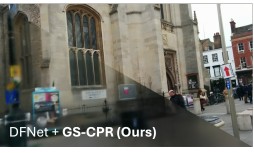

(a) 7Scenes      (b) Cambridge

Figure 6: A comparison between DFNet + GS-CPR and DFNet + NeFeS$_{50}$.

## 5 CONCLUSION

We present GS-CPR, a novel test-time camera pose refinement framework leveraging 3DGS for scene representation to improve the localization accuracy of state-of-the-art APR and SCR methods. GS-CPR enables one-shot pose refinement using only a single RGB query and a coarse initial pose estimate from APR and SCR methods. Our approach outperforms existing NeRF-based optimization methods in both accuracy and runtime across various indoor and outdoor visual localization benchmarks, achieving new state-of-the-art accuracy on two indoor datasets. These results demonstrate the effectiveness and efficiency of our proposed framework.

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

# A APPENDIX

## A.1 GT POSES DETAILS

In Section 4.2, we report evaluation results based on the SfM ground truth (GT) poses for the 7Scenes dataset, as these poses can render higher quality images (Chen et al., 2024a). Since NeFeS (Chen et al., 2024a) demonstrates the superior accuracy of SfM poses using NeRF as the scene representation, we provide a quantitative comparison in Table 9 and illustrative rendering examples of 3DGS in Figure 7. These results affirm that SfM poses are more accurate, leading to higher quality rendered images and depth maps when using 3DGS. We utilize pre-built COLMAP models from Brachmann et al. (2021) for 7Scenes and 12Scenes datasets, and the models from HLoc toolbox (Sarlin et al., 2019) for the Cambridge landmarks dataset. For the 7Scenes dataset, we enhance the accuracy of the sparse point cloud by utilizing dense depth maps provided by the dataset, combined with the HLoc toolbox and rendered depth maps (Brachmann & Rother, 2021).

Table 9: Quatitative comparison between the 3DGS models implemented in Section 4.1 trained by dSLAM GT poses and SfM GT poses. We report the average PSNR (dB) for the test frames in each scene. The best results are in bold (higher is better).

|  | dSLAM GT | SfM GT |
| --- | --- | --- |
| Scenes | avg. PSNR ↑ | avg. PSNR ↑ |
| chess | 19.6 | **23.1** |
| fire | 19.8 | **21.2** |
| heads | 18.4 | **19.7** |
| office | 19.4 | **21.7** |
| pumpkin | 20.3 | **23.2** |
| redkitchen | 18.5 | **21.4** |
| stairs | 19.7 | **20.1** |
| avg. | 19.4 | **21.5** |

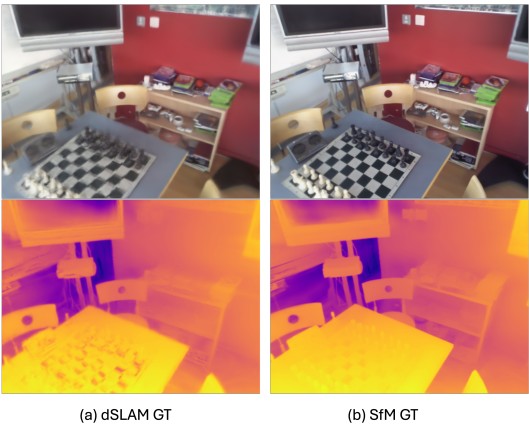

(a) dSLAM GT        (b) SfM GT

Figure 7: Render performance example (dSLAM GT vs. SfM GT). The 3DGS model trained with SfM GT poses (b) renders superior geometric details compared to the dSLAM 3DGS (a) for the same query image, particularly in the chessboard and pieces area.

## A.2 SEMANTIC SEGMENTATION WHEN BUILDING 3DGS

To handle challenges in outdoor datasets, we apply temporal object filtering to filter out moving objects in the dynamic scene using an off-the-shelf method (Cheng et al., 2022), leading to better accurate scene reconstruction quality and pixel-matching performance. We show examples of semantic segmentation in Figure 8 and its effect on novel view synthesis (NVS) results in Figure 9. This approach, together with ACT, allows our 3DGS models to provide more robust and better rendering results.

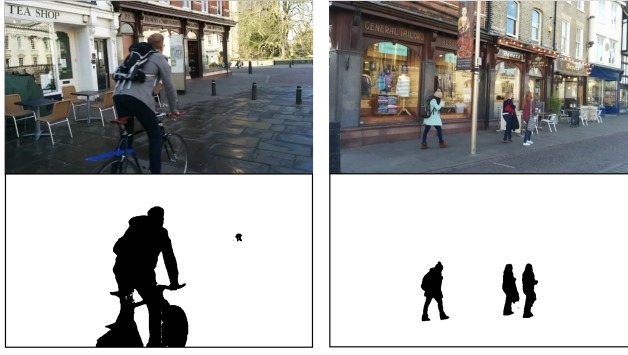

Figure 8: Example of masking on the ShopFacade scene. Top: original images; Bottom: corresponding semantic masks.

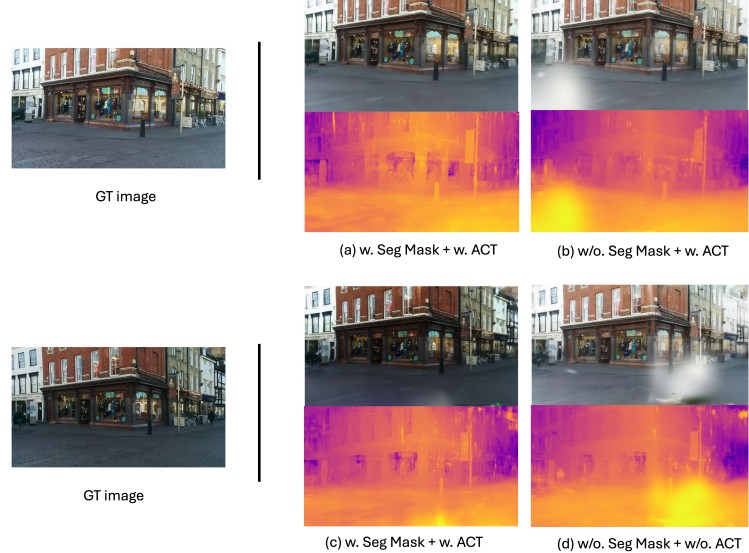

Figure 9: Rendering performance comparison. The 3DGS model trained with segmentation masks renders superior geometric details and fewer artifacts compared to the model trained without masks.

A.3    THE ADVANTAGES OF GS-CPR OVER OTHER APPROACHES

**Advantages over render and comapre methods:** Methods (Yen-Chen et al., 2021; Lin et al., 2023; Chen et al., 2024a; Sun et al., 2023; Trivigno et al., 2024) leverage only the geometric information of the representation for rendering but do not use it for 2D-3D matching. Consequently, they offer limited accuracy gains and are hindered by slow convergence and high computational costs due to iterative rendering. While NeFeS (Chen et al., 2024a) reduces rendering time and cost by using feature maps and feature loss rather than photometric loss, its accuracy potential remains lower than methods employing 2D-3D matches from original RGB images due to the loss of information in feature maps.

**Advantages over structure-based methods:**    Classical 3D structure-based methods, such as HLoc (Dusmanu et al., 2019; Sarlin et al., 2019; Taira et al., 2018; Noh et al., 2017; Sattler et al., 2016; Sarlin et al., 2020; Lindenberger et al., 2023), estimate camera poses using a 3D SfM point cloud and a reference image database. HLoc requires storing a descriptor database and retrieving the top-$k$ most similar images for 2D-3D correspondences, typically requiring $k$= 5 to 40 images for robust localization (Humenberger et al., 2022; Sarlin et al., 2022; Leroy et al., 2024). Our approach offers two key advantages: (1) While HLoc requires $k$ matching operations, our GS-CPR only requires one, and its single-shot pose optimization surpasses the accuracy of traditional HLoc. (2) For

challenging queries, even the top-1 retrieved image may have limited overlap with the query (Liu et al., 2024b). However, since GS-CPR performs NVS based on APR and SCR predictions, the rendered images exhibit a greater overlapping region with the query, leading to more accurate matches. We provide examples in Figure 10. The key insight is that both image retrieval and ACE pose-based retrieval are restricted to identifying queries within a limited reference pool. In contrast, our approach theoretically allows for an unlimited reference pool. (3) Using 3DGS instead of sparse point clouds for scene representation enables the domain shift of the rendered image according to the query's exposure through a learning approach, offering greater flexibility.

**System design analysis:** Our approach goes beyond simply combining 3DGS and MASt3R. As outlined in Section 3.2, our method leverages the matching components of MASt3R to eliminate the need for training extra features to match image pairs with a sim-to-real domain gap—a common limitation of other NeRF-based pose estimation techniques. However, relying solely on MASt3R with reference images fails to deliver accurate metric translation due to the lack of scale information and cannot build 2D-3D matches for absolute pose estimation. This limitation arises because MASt3R is unable to generate metric 3D points within the pre-built global coordinate system. For instance, Jiao et al. (2024) addresses this problem in robotics tasks by incorporating a depth camera. To resolve this challenge, 3DGS in our framework serves a critical function by rendering metric depth, enabling accurate 2D-3D matching. Besides, the rendered view generated by 3DGS from SCR and APR poses aligns much better than normal image retrieval from fixed reference images. This integration is important in recovering precise scale and achieving robust and accurate pose estimation with sufficient matches. By combining the strengths of these components, our framework addresses current limitations.

## A.4 SUPPLEMENTARY VISUALIZATION

To complement our quantitative analysis, we present additional results in Figure 11 that provide a qualitative perspective on pixel-wise alignment using NVS based on 3DGS across three datasets. A video is also included in the supplementary material.

## A.5 FAILURE CASES AND LIMITATION

One limitation of our method lies in its dependency on the accuracy of the initial pose estimates provided by the pose estimator. When the initial pose is highly inaccurate, the overlap between the rendered images and the query image is insufficient to establish reliable 2D-3D correspondences for accurate pose estimation. As shown in Figure 12, GS-CPR cannot refine the DFNet's initial pose in this case because it is too far away from the GT pose.

Following Section 4.5 of NeFeS (Chen et al., 2024a), we conduct quantitative experiments to evaluate the limitations of GS-CPR. Specifically, we introduce random perturbations to the ground truth poses of test frames on the ShopFacade scene, applying fixed magnitudes of rotational and translational errors independently. The results after pose refinement using GS-CPR are presented in Table 10 and Table 11. Our framework can improve the accuracy when rotation error $< 50°$ and translation error $< 8$ meters, respectively. In contrast, NeFeS achieves accuracy improvements only for rotational errors under $35°$ and translational errors below $4$ meters. These findings highlight that our method significantly expands the optimization range, enhancing its robustness to larger pose perturbations.

Table 10: Average rotation error after refinement by GS-CPR.

| Jitter-magnitude (°) | 5 | 10 | 20 | 30 | 40 | 50 | 55 | 60 |
|---|---|---|---|---|---|---|---|---|
| Avg. Rot. Error (°) | 0.23 | 0.23 | 0.27 | 0.35 | 0.6 | 7 | 26 | 83 |

Table 11: Average translation error after refinement by GS-CPR.

| Jitter-magnitude (m) | 1 | 2 | 3 | 4 | 5 | 6 | 8 | 10 |
|---|---|---|---|---|---|---|---|---|
| Avg. Trans. Error (m) | 0.19 | 0.38 | 0.51 | 0.88 | 1.13 | 2.0 | 3.1 | 10.7 |

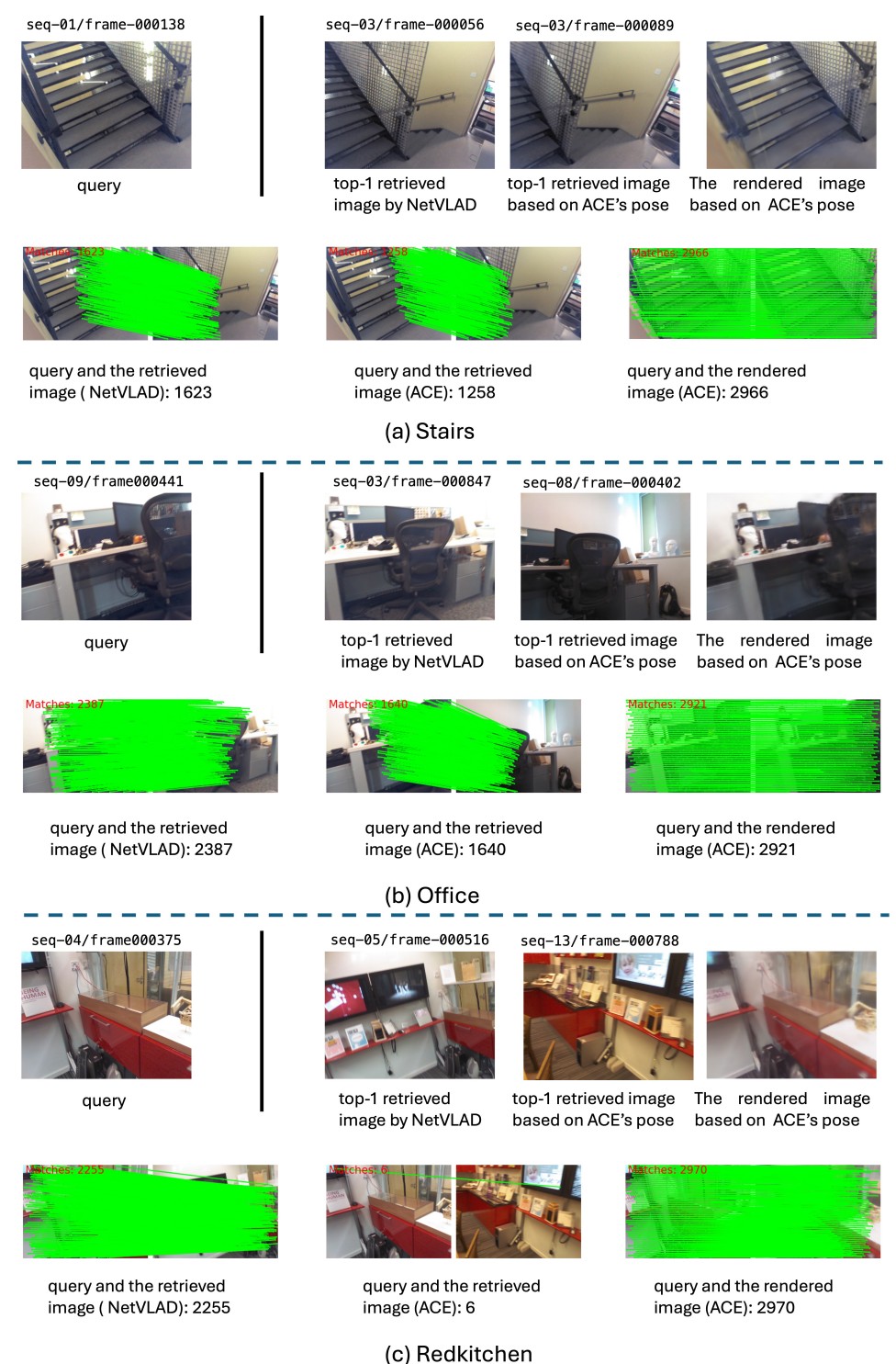

Figure 10: The image rendered from the pose estimator's predictions exhibits a greater overlapping region with the query image than the one retrieved by NetVLAD (Arandjelovic et al., 2016) and the one retrieved by ACE's pose. We use MASt3R as the matcher. Since the matches are very dense, we show the number of matches but only visualize 20% of the matches.

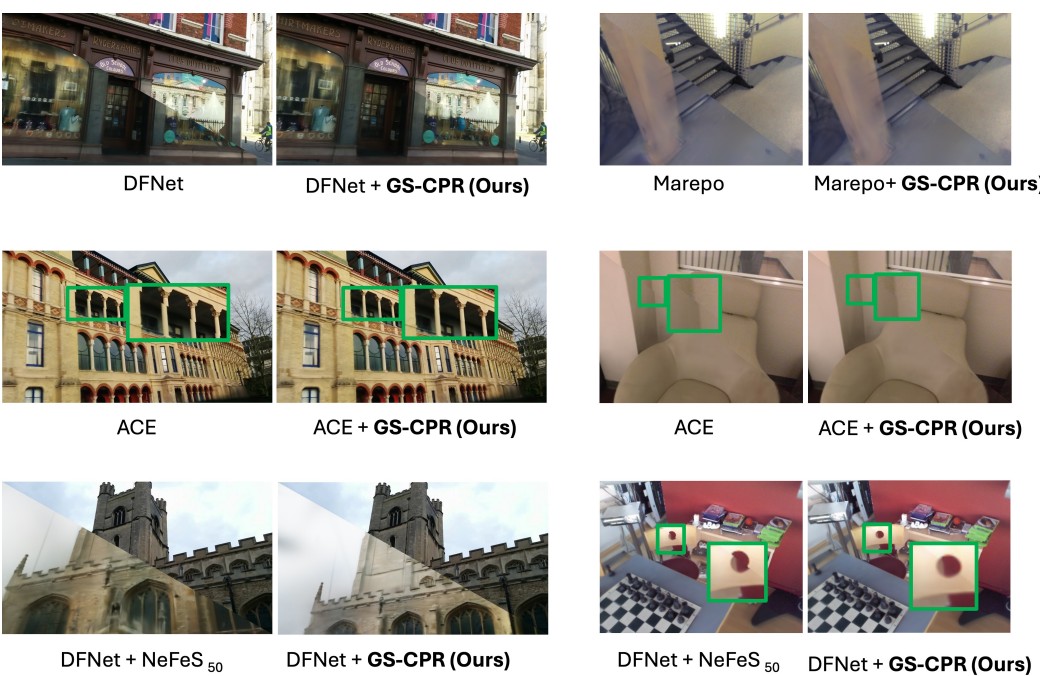

Figure 11: Each subfigure is divided by a diagonal line, with the **bottom left** part rendered using the estimated/refined pose and the **top right** part displaying the ground truth image. Patches highlighting visual differences are emphasized with green insets for enhanced visibility.

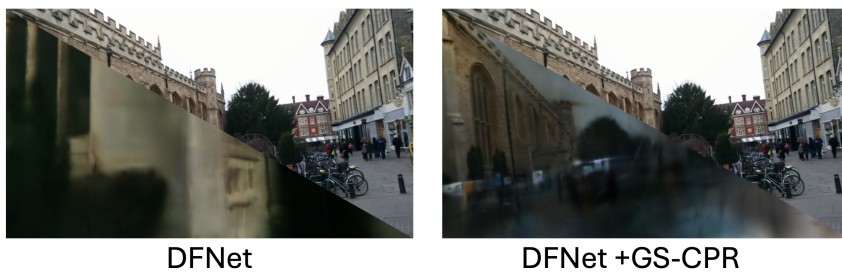

Figure 12: Failure case example. Each subfigure is divided by a diagonal line, with the **bottom left** part rendered using the estimated/refined pose and the **top right** part displaying the ground truth image.

This paper demonstrates the effectiveness of our framework on commonly used datasets and benchmarks. However, reconstructing high-quality 3DGS models for large scenes remains a significant challenge. Exploring the application of this framework to large-scale scenes for accurate visual camera relocalization is a promising avenue for future work.

Table 12: We report the average accuracy (%) of frames meeting a $[5\text{cm}, 5°]$, $[2\text{cm}, 2°]$ pose error threshold, and the median translation and rotation errors (cm/°) across 7Scenes and 12Scenes.

| Datasets | Methods | Avg. Err ↓ [cm/°] | Avg. ↑ [5cm, 5°] | Avg. ↑ [2cm, 2°] |
|---|---|---|---|---|
| 7Scenes | GLACE | 1.2/0.36 | 95.6 | 82.2 |
| | GLACE + **GS-CPR (ours)** | **0.8/0.27** | **99.5** | **90.7** |
| 12Scenes | GLACE | 0.7/0.25 | **100** | 97.5 |
| | GLACE + **GS-CPR (ours)** | **0.5/0.21** | **100** | **98.9** |

Table 13: Comparisons on Cambridge Landmarks dataset. We report the median translation and rotation errors (cm/°) of different methods.

| Methods | Kings | Hospital | Shop | Church | Avg. ↓ [cm/°] |
|---|---|---|---|---|---|
| GLACE[1] | 20/0.32 | 20/0.41 | 5/0.22 | 9/0.3 | 14/0.32 |
| GLACE[2] | 19/0.3 | **17**/0.4 | **4/0.2** | 9/0.3 | **12**/0.3 |
| GLACE + **GS-CPR (ours)** | 17/0.28 | 18/**0.34** | 5/0.21 | **8/0.28** | **12/0.28** |

[1] Accuracy based on official open-source models (Wang et al., 2024a).
[2] Accuracy reported in the paper (Wang et al., 2024a).

## A.6 SUPPLEMENTARY EXPERIMENTS

GLACE (Wang et al., 2024a) is an enhanced version of ACE tailored for large-scale outdoor scenes, while exhibiting nearly identical accuracy in indoor environments compared to ACE. We present the results of GLACE + GS-CPR in Tables 12 and 13 to provide supplementary results for evaluating the performance of our approach. GS-CPR significantly improves GLACE accuracies in two of the three datasets (7scenes and 12scenes), demonstrating the effectiveness of our method. On the Cambridge Landmarks dataset, we achieve comparable results, with a slight edge in rotational error.

