# OpenReview forum: "GS-CPR: Efficient Camera Pose Refinement via 3D Gaussian Splatting"
_ICLR.cc/2025/Conference — ICLR 2025 Poster_

### Official Review · Reviewer_UwQs · 2024-10-30

**Soundness:** 3
**Presentation:** 3
**Contribution:** 3
**Rating:** 6
**Confidence:** 4

**Summary:**

This proposes a novel visual localization method that utilizes 3D Gaussian Splatting. This approach replaces the point cloud in the pipeline of visual localization with a 3DGS model as the 3D representation. The camera pose of a query image is computed in a coarse-to-fine manner. The coarse pose is predicted by using a pretrained pose estimator. The pose refinement is carried out by neural rendering and image matching. The method is evaluated across indoor and outdoor datasets, outperforming state-of-the-art NeRF-based methods and traditional localization methods.

**Strengths:**

The idea of using 3DGS for visual localization is simple yet effective. It replaces the key components in the localization pipeline with advanced alternatives. For instance, the image retrieval module is replaced with a pose estimator for a coarse pose prediction. The 2D-3D matching is performed based on a prebuilt 3DGS model instead of the point cloud. The authors take the challenges, such as the appearance gap, into account and introduce an exposure adaption module, which enhances the appearance consistencies. The proposed method achieves more accurate camera pose estimates than the existing localization approaches on several datasets. The paper is well-written and easy to follow.

**Weaknesses:**

-	The superiority of 3DGS in the framework of visual localization is a bit unclear. The scalability of 3DGS seems questionable. For instance, in a large-scale and unbounded scenario, the 3D map might be built from thousands of images in the traditional localization pipeline. The 3DGS appears to be inapplicable in this case. If this is the case, the requirement of the 3DGS model might be a limitation in real applications.

-	The authors argue that in HLoc, the top-1 retrieved image may have limited overlap with the query. However, this is the limitation of the current image retrieval techniques rather than a shortcoming of HLoc itself. In the introduced pipeline, the coarse pose is computed using an off-the-shelf approach, which might be more powerful than image retrieval. Therefore, this approach could also benefit HLoc. For example, the reference can be selected based on the predicted camera pose. The advantage of 3DGS is not convincing enough.

-	The evaluation metrics are inconsistent across different datasets. The authors report accuracy in some cases and pose error in others.

-	The comparison with traditional localization methods like HLoc on Cambridge Landmarks is missing.

**Questions:**

What is the major difference compared with 3DGS SLAM methods since they can also optimize the camera poses?

---

> ### Author Response · Authors · 2024-11-20
> **Response to Reviewer UwQs (1 out of 2)**
>
> Thank you for your positive comments, including the recognition that the proposed method is simple yet effective and achieves more accurate results than state-of-the-art NeRF-based methods. Below, we address your concerns and respond to your questions.
>
> -(a) **Scalability**: We think the large-scale and unbounded scene reconstruction is an open problem in the community. Most of the structure-based methods that rely on scene geometry have similar problems under large-scale scene relocalization. However, visual relocalization methods that do not rely on scene representation have better scalability but much lower accuracy [1]. Currently, several concurrent studies [2][3][4] are exploring large-scale 3DGS reconstruction. These advancements hold the potential to be integrated into our existing framework in the future. In this paper, we demonstrate the high performance of our approach across three widely used public datasets, highlighting its effectiveness. We’d love to further explore this challenging scenario in our future works.
>
> -(b) **Advantages of 3DGS in the relocalization task**. Our framework leverages novel view synthesis (NVS) from 3DGS to render images that provide more robust and sufficient 2D-3D correspondences for RANSAC + PnP, outperforming traditional reference image matching, particularly in scenarios with sparse reference databases. Unlike the traditional visual localization pipeline, which is constrained to a fixed set of predefined reference images, our 3DGS-based NVS can generate ANY view based on the pose estimator's predictions. By utilizing 3DGS instead of sparse point clouds for scene representation, our method employs a learning-based approach to adapt the domain of rendered images to the query's exposure conditions, ensuring closer alignment with the query and offering greater flexibility.
>
> -(c) **Compared with HLoc**: HLoc still has a marginal advantage regarding translation accuracy on the Cambridge dataset. We updated the results of HLoc (SP + SG) on Cambridge landmarks in Table 3 in the rebuttal version. However, this paper focuses on comparisons within the state-of-the-art neural render pose (NRP) estimation approaches.  We conduct experiments on the 7Scenes dataset using the open-source HLoc toolbox [5] with SfM-generated ground truth (GT) to compare our approach against two baselines: HLoc (SP + SG) + NetVLAD (k=1) and ACE + HLoc (SP + SG) with nearest pose-based retrieval (k=1).
>
> | Methods | Avg. median error (cm/$^\circ$) $\downarrow$  |
> |-----------|----------|
> |NetVLAD + HLoc (SP + SG) |6/2.2|
> |ACE + HLoc (SP + SG) |7/2.4|
> |ACE + GSLoc |**0.8/0.25**|
>
> The results show that pose-based image retrieval from ACE cannot get better results and our ACE + GSLoc is more accurate.
>
> -(d) **Evaluation metrics**: We have supplemented the evaluation with both accuracy and pose errors for the 12Scenes dataset and updated Table 4 accordingly. For the Cambridge Landmarks dataset, we report only pose errors to ensure a consistent and fair comparison, as previous studies provide only this metric.
>
> | Methods                | Avg. Err ↓ [cm/°] | Avg. ↑ [5cm, 5°] | Avg. ↑ [2cm, 2°] |
> |------------------------|-------------------|-------------------|-------------------|
> | DSAC*                 | **0.5** / 0.25       | 99.8             | 96.7             |
> | Marepo                | 2.1 / 1.04       | 95               | 50.4             |
> | ACE                   | 0.7 / 0.26       | **100**           | 97.2             |
> | GLACE                 | 0.7 / 0.25       | **100**              | 97.5             |
> | Marepo + GSLoc| 0.7 / 0.28       | 98.9             | 90.8             |
> | ACE + GSLoc | **0.5 / 0.21**       | **100**             | 98.7             |
> | GLACE + GSLoc| **0.5 / 0.21**       | **100**              | **98.9**             |
>
> -(e) **Differences between 3DGS SLAM**: We believe our framework and 3DGS SLAM are solving different problems, please refer to our response to reviewer tSXc (a) for more details. In addition, for pose refinement, current popular 3DGS SLAM systems primarily leverage photometric loss to optimize both the map and the camera's trajectory iteratively with differentiable rendering, which is more similar to iNeRF and iComMa. However, our framework enables one-shot absolute pose refinement.
>
> [1] Sattler, Torsten, et al. "Understanding the limitations of cnn-based absolute camera pose regression." CVPR 2019.
>
> [2] Yan, Yunzhi, et al. "Street gaussians for modeling dynamic urban scenes." ECCV 2024
>
> [3] Liu, Yang, et al. "Citygaussian: Real-time high-quality large-scale scene rendering with gaussians." ECCV 2024
>
> [4] Liu, Xi, et al. "3dgs-enhancer: Enhancing unbounded 3d gaussian splatting with view-consistent 2d diffusion priors." arXiv 2024
>
> [5] Sarlin, Paul-Edouard, et al. "From coarse to fine: Robust hierarchical localization at large scale." CVPR 2019.

---

> > ### Comment · Reviewer_UwQs · 2024-11-25
> >
> > I appreciate the authors' response and the additional results. As the authors acknowledge the scalability issue, I believe it would be beneficial to explicitly mention this as a limitation in the main paper. I also commend the inclusion of further comparisons with HLoc in the main paper, which provides a comprehensive understanding of both the proposed method and HLoc.
> >
> > However, I am a bit unclear about the ablation on image retrieval. Could you kindly clarify why HLoc is not compatible with ACE?

---

> ### Author Response · Authors · 2024-11-22
> **Response to Reviewer UwQs (2 out of 2)**
>
> -(f) **More visualization results**: In the new rebuttal version, we provide match visualizations of multiple scenes as well as examples of retrieved images in Figure 10. This would better complement the results in (c) **Compared with HLoc**. We also give you detailed information about each image, which will help you reproduce the results easily. These examples illustrate the advantages of using 3DGS render images from SCR methods’ prediction to perform matching. The key insight is that both image retrieval and ACE pose-based retrieval are restricted to identifying queries within a limited reference pool. In contrast, our approach theoretically allows for an unlimited reference pool.

---

> ### Author Response · Authors · 2024-11-25
> **Response to Reviewer UwQs**
>
> Thank you for your follow-up questions. We appreciate the opportunity to address your concerns.
>
> While we acknowledge that 3DGS has limitations regarding large-scale scalability, we view this as an open problem within the field. Additionally, we compare HLoc in Tables 2 and 3 of this paper, but our primary focus of this paper is comparing with NeRF-based methods, such as NeFeS and CrossFire. If scalability is considered a significant limitation, it is worth noting that all previous neural render pose (NRP) estimation methods share similar constraints. Large-scale scenes also pose challenges for traditional HLoc, which relies on the SfM sparse point cloud model.
>
> **Why is HLOC not compatible with ACE?** We never state or show that HLoc is not compatible with ACE. We just show that ACE cannot get better-retrieved images than traditional image retrieval methods because they can only retrieve images from the same pool. The key insight is that both image retrieval and ACE pose-based retrieval are restricted to identifying queries within a limited reference pool. In contrast, our approach theoretically enables an unlimited reference pool by leveraging high-quality rendered images from 3DGS.

---

> > ### Comment · Reviewer_UwQs · 2024-11-25
> >
> > Thank you for providing further clarification. To clarify my earlier point, I was not criticizing scalability as a weakness of this paper. Rather, since scalability is indeed a critical challenge in visual localization, I was suggesting that it might be worth mentioning this in the main paper. Doing so could inspire new directions for future research and follow-up work.
> >
> > Regarding my second question about ACE, I realize there was a misunderstanding on my part. The explanation you provided has now made it clear.

---

> ### Author Response · Authors · 2024-11-25
> **Response to Reviewer UwQs**
>
> Thank you very much for your response. We will add a description in the limitation section like "This paper demonstrates the effectiveness of our framework on commonly used datasets and benchmarks. However, reconstructing high-quality 3DGS models for large scenes remains a significant challenge. Exploring the application of this framework to large-scale scenes for accurate visual camera relocalization is a promising avenue for future work." in the final version.

---

### Official Review · Reviewer_tSXc · 2024-10-31

**Soundness:** 3
**Presentation:** 3
**Contribution:** 3
**Rating:** 6
**Confidence:** 4

**Summary:**

The paper pays attention on camera pose refinement with 3D Gaussian Splatting as a scene representation, which aims to enhance the localization accuracy of state-of-the-art absolute pose regression and scene coordinate regression methods. A refinement framework, called GSLoc, is proposed to establish 2D-3D correspondences by an exposure-adaptive module. Experiments demonstrate that GSLoc improves the localization accuracy over different relocaliztion methods.

**Strengths:**

1 The paper proposes a novel test-time pose refinement framework, termed GSLoc, which employs 3D Gaussian Splatting for scene representation and utilizes the 3D vision foundation model MASt3R for precise matching.

2 GSLoc enhances the pose estimation accuracy of both APR and SCR methods across these benchmarks.

**Weaknesses:**

1 The main contribution is incremental.

(1)Firstly, the motivation of GSLoc is less interesting. From my viewpoint, the usage of feature matching for pose refinement is not novel. Meanwhile, the paper just uses 3D Gaussian Splatting for scene representation instead of traditional hand-crafted point cloud. Compared with Gaussian Splatting SLAM systems (like Gaussian SLAM [1, 2]), the idea of this paper is less innovative.

(2)Secondly, the innovative contributions are few. The paper seems a combination of current hot technologies, like 3D Gaussian Splatting and the 3D vision foundation model MASt3R.

[1]Gaussian Splatting SLAM, CVPR 2024

[2]GS-SLAM: Dense Visual SLAM with 3D Gaussian Splatting

[3]LoFTR: Detector-Free Local Feature Matching with Transformers, CVPR 2021

[4]DKM: Dense Kernelized Feature Matching for Geometry Estimation

2 The localization accuracy comparisons with feature matching based methods are missing. The core idea behind GSLoc is feature matching with 3D Gaussian Splatting. It is curious that whether GSLoc is also effective for pose refinement of leaned feature matching based methods, like LoFRT[3], DKM[4] or others. It is suggested to add these results.


3 The overlap of Figure 2 and 3 is somwhat high. It is suggested to modify Figure 3 to express the difference of GSLocrel in comparison with GSLoc.

**Questions:**

The question and suggestions are listed in the part of Weaknesses.

---

> ### Author Response · Authors · 2024-11-20
> **Response to Reviewer tSXc (1 out of 2)**
>
> Thank you for your positive comments on our evaluation results. We primarily address your concerns below.
>
> -(a) **Difference between GS-SLAM**: **SLAM and camera relocalization are fundamentally different tasks with distinct objectives.** Compared to GS-SLAM, our framework focuses on determining the pose of a single RGB query image relative to a pre-built map without relying on any prior frame information.  SLAM involves concurrently building a map of an unknown environment and estimating the camera's motion across sequential frames, assuming smooth input and a continuous camera trajectory. Consequently, camera poses are determined relative to previous frames. However, this causes issues such as drift. Popular GS-SLAM systems primarily address this tracking challenge by leveraging photometric loss to optimize both the map and the camera's trajectory iteratively like iNeRF [1] and iComMa [2] with differentiable rendering. However, our framework enables one-shot absolute pose estimation.
>
> Unlike SLAM, where the map evolves alongside the camera trajectory, our task aims to estimate the camera pose within a global pre-established map. Our framework is explicitly designed for the (re)localization task, achieving more accurate results than other state-of-the-art neural render pose (NRP) estimation pipelines.
>
> **Table 1. We list the main differences between SLAM and Relocalization.**
> | **Aspect**                    | SLAM (3DGS SLAM)                                                                                 | Relocalizer (GSLoc)                                                                               |
> |--------------------------------|---------------------------------------------------------------------------------------------|-------------------------------------------------------------------------------------------|
> | **Task Objective**             | Simultaneous Localization and Mapping (SLAM) - build a map and estimate sequential poses.  | Camera (Re)localization - determine the absolute pose of a single RGB query image relative to a global map.   |
> | **Input Dependency**           | Requires sequential frames with smooth input and continuous camera trajectory.             | Operates on a single RGB image without prior frame information.                          |
> | **Map Type**                   | Building a map of an unknown environment                                       | Pre-built  map.                                                                    |
> | **Pose Estimation**            | Relies on relative poses between consecutive frames, causing accumulated drift.               | Estimates absolute pose directly within the static map.                                  |
>
> -(b) **Novelty**: The scope of this paper is focused on comparisons within the state-of-the-art NRP estimation approaches such as NeFeS, CrossFire, and other pose-established methods like SCR and APR. Existing NeRF-based approaches typically rely on training scene-specific descriptors, as seen in methods like CrossFire and NeFeS. In contrast, our framework eliminates the dependence on extra feature training through the integration of MASt3R, while simultaneously reducing the domain gap between rendered images and query images by introducing the ACT module in 3DGS. Using 3DGS instead of sparse point clouds for scene representation enables the domain shift of the rendered image according to the query's exposure through a learning approach, offering greater flexibility. As demonstrated in Table 8, Figure 5, and Figure 9, the off-the-shelf Scaffold-GS approach alone does not achieve optimal results. By incorporating ACT and semantic filtering during 3DGS training, we significantly improve rendering quality and achieve superior performance. These modules make our framework easier to deploy and achieve higher accuracy than other NRP methods.
>
> [1] Yen-Chen, Lin, et al. "inerf: Inverting neural radiance fields for pose estimation." IROS 2021
>
> [2] Sun, Yuan, et al. "icomma: Inverting 3d gaussians splatting for camera pose estimation via comparing and matching." arXiv 2023

---

> ### Author Response · Authors · 2024-11-20
> **Response to Reviewer tSXc (2 out of 2)**
>
> -(c) **Combine hot approaches**: Our approach goes beyond simply combining 3DGS and MASt3R. As outlined in Section 3.2, our method leverages the matching components of MASt3R to eliminate the need for training extra feature descriptors to match image pairs with a sim-to-real domain gap—a common limitation of other NeRF-based pose estimation techniques. However, relying solely on MASt3R with reference images fails to deliver accurate metric translation due to the lack of scale information and cannot build 2D-3D matches for absolute pose estimation. This limitation arises because MASt3R is unable to generate metric 3D points within the pre-built global coordinate system. For instance, [1] addresses this problem in robotics tasks by incorporating a depth camera.
> To resolve this challenge, 3DGS serves a critical function by rendering metric depth and constructing 3D geometry, enabling accurate 2D-3D matching. Besides, the rendered view generated by 3DGS from SCR/APR poses aligns much better than normal image retrieval from fixed reference images. This integration is important in recovering precise scale and achieving robust and accurate pose estimation with sufficient matches. By combining the strengths of these components, our framework addresses current limitations.
>
> -(d) **Feature matching comparison**: We compare three different matching methods in Table 7 and discuss this in Section 4.4. That shows that MASt3R has higher accuracy than LoFTR and DUSt3R.
>
> -(e) **Figure 3 modification**: We appreciate your feedback and revised Figure 3 to more clearly highlight the distinction between GSLoc_rel and GSLoc.
>
> [1] Jiao, Jianhao, et al. "LiteVLoc: Map-Lite Visual Localization for Image Goal Navigation." arXiv 2024.

---

> > ### Comment · Reviewer_tSXc · 2024-11-26
> >
> > Thank you for your response. The response has successfully addressed most of my concerns, and I could increase my score from 5 to 6.

---

### Official Review · Reviewer_HMwQ · 2024-11-02

**Soundness:** 3
**Presentation:** 4
**Contribution:** 2
**Rating:** 6
**Confidence:** 3

**Summary:**

This paper proposes a new approach to optimize the initial camera pose in visual localization. Using 3DGS as the scene representation, RGB images and depth maps are rendered based on the initial camera pose of the query image estimated by other methods. Then, 2D-2D matches between the query image and the rendered results are computed via MASt3R and elevated to 2D-3D matches using the depth maps. Finally, the relative pose is solved through PnP to optimize the initial camera pose. Compared to previous camera pose optimization methods based on cost functions and optimization, the GSLoc pipeline achieves one-shot pose optimization, offering advantages in both speed and accuracy. The effectiveness of GSLoc has been validated on indoor and outdoor visual localization datasets.

**Strengths:**

- The GSLoc pipeline combines the advantages of speed and accuracy. Through high-quality 3DGS reconstruction and rendering, it directly calculates optimized camera poses using image matching, which enhances the robustness and usability of the system compared to optimization methods based on specific cost functions.
- The writing of the paper is easy to follow.
- Comprehensive experiments have been conducted, demonstrating advantages over other camera pose optimization methods and traditional visual localization approaches.

**Weaknesses:**

- The paper lacks certain experiments to demonstrate the necessity of the proposed method:
    - Table 2 shows the effectiveness of GSLoc compared to HLoc. However, it does not pair HLoc with an improved initial pose estimation method, such as ACE, as done for GSLoc. A reasonable baseline would be to use ACE to estimate the initial camera pose, retrieve the top-K images with the closest poses from the database, and then compute the localization result using HLoc and optionally refinet it with PixLoc.
- The novelty of the method is limited, as the effectiveness of the pipeline primarily relies on improvements in reconstruction and rendering quality provided by Scaffold-GS as the scene representation.
- There is a lack of analysis on the method’s applicability:
    - Feasibility of high-quality 3DGS reconstruction: Reconstructing high-quality 3DGS to render the images needed for GSLoc requires densely sampled viewpoints. Moreover, modeling requires images with viewpoints very close to that of the query. How robust is GSLoc when there are no views close to the query viewpoint (i.e., in cases of wide baselines)?
    - Impact of significant appearance changes on the method (e.g., on datasets like Aachen-day-night or similar ones of smaller scenes).
    - Feasibility of the method for large-scale visual localization (e.g., Aachen-day-night).

**Questions:**

- What are the feature detector and matcher used by HLoc in the comparison shown in Table 2?
- In the runtime analysis, there seem to be some issues with the time calculation for GSLoc and GSLoc_rel.

---

> ### Author Response · Authors · 2024-11-20
> **Response to Reviewer HMwQ**
>
> Thank you very much for your positive feedback on our paper, acknowledging the novelty of our approach, advantageous performance, excellent presentation, and good writing. We primarily address your concerns below.
>
> -(a) **Necessity**: Enhanced Matching through Neural View Synthesis (NVS): Our framework uses NVS to render images that provide
> matches for RANSAC + PnP.  Unlike HLoc, which is limited to a predefined set of reference images, 3DGS-based NVS is not constrained by the reference images database, instead rendering virtually unlimited views. This is especially beneficial when the reference database is sparse. Our framework can thus ensure closer alignment with the query. We conduct experiments on the 7Scenes dataset using the open-source HLoc toolbox [1] with SfM-generated ground truth (GT) to compare our approach against two baselines: HLoc (SP + SG) + NetVLAD (k=1) and ACE + HLoc (SP + SG) with nearest pose-based retrieval (k=1). The parameter k=1 is selected to ensure a fair comparison, as our framework is designed to match a single rendered image with the query.
>
> | Methods | Avg. median error (cm/$^\circ$) $\downarrow$  |
> |-----------|----------|
> |NetVLAD + HLoc (SP + SG) |6/2.2|
> |ACE + HLoc (SP + SG) |7/2.4|
> |ACE + GSLoc |**0.8/0.25**|
>
> The results show that pose-based image retrieval from ACE cannot get better results than our ACE + GSLoc.
>
> -(b) **Novelty**: The significant performance improvements obtained by GSLoc, compared to other neural render pose (NRP) estimation methods were not directly from off-the-shelf Scaffold-GS, which yields suboptimal results, as shown in Table 8, Figure 5, and Figure 9 of our paper. We significantly enhance the rendering quality for localization by incorporating ACT and semantic filtering during 3DGS training. Additionally, MASt3R eliminates the necessity of training scene-specific features. The integration of these modules collectively leads to substantial improvements over state-of-the-art NRP estimation pipelines, highlighting the effectiveness and flexibility of our proposed approach.
>
> -(c) **3DGS Rely on Dense View**: We thank the reviewer for mentioning the issue, we perform experiments without using dense view 3DGS reconstruction. To evaluate the robustness of our approach under sparse view conditions, we uniformly sample 1/3, 1/10, and 1/20 of the training images (2000 frames) on Stairs of 7Senes to construct the 3DGS model and evaluate localization accuracy on this reduced dataset. We provide results here, demonstrating the effectiveness and robustness of our method even with sparse training views.
>
> |     |DFNet   |+ GSLoc (1/20) |  + GSLoc (1/10) |  + GSLoc (1/3)|   + GSLoc (full)|
> |-----------|----------|-----------|----------|----------|----------|
> |5cm/5$^\circ$(%)($\uparrow$)|         9.1   |          80.1    |             81.5       |         90.0		|  90.8      |
>
>
> -(d) **Aachen-day-night**: we’d love to further explore this challenging scenario in our future works. Our intuition about current 3DGS/NeRF is that they cannot handle the day-night gap very well. Several concurrent studies [2][3][4] are exploring large-scale reconstruction and lighting changes for 3DGS. These advancements hold the potential to be integrated into our existing framework in the future. However, we hope our strong results in commonly used datasets are enough to show the advantages over existing NRP methods.
>
> -(e) **HLoc setting in Table 2**: We report the HLoc results using DenseVLAD as the image retrieval module (k=20), combined with SuperPoint and SuperGlue, based on the official open-source implementation provided in [5].
>
> -(f) **Runtime**: We rely on SCR/APR methods to provide initial poses, enabling fast inference.  We provide runtime clarification of GSLoc and GSLoc_rel on RTX 4090 below.
>
> GSLoc: 3DGS rendering (10ms) + MASt3R inference (70ms) + MASt3R matching (40ms)+ RANSAC+PnP (50ms) ~ 170ms
>
> GSLoc_rel: 3DGS rendering (10ms) + MASt3R inference (70ms) ~ 80ms
>
> Total runtime:
> SCR + GSLoc: ACE inference (17ms) + 170ms
>
> APR + GSLoc: Marepo inference (7.4ms) + 170ms;  DFNet inference (2ms) + 170ms
>
> SCR + GSLoc_rel: ACE inference (17ms) + 80ms
>
> APR + GSLoc_rel: Marepo inference (7.4ms) + 80ms;  DFNet inference (2ms) + 80ms
>
> Thanks for catching the runtime calculation issue, we will update the runtime data in Table 6.
>
> [1] Sarlin, Paul-Edouard, et al. "From coarse to fine: Robust hierarchical localization at large scale." CVPR 2019.
>
> [2] Yan, Yunzhi, et al. "Street gaussians for modeling dynamic urban scenes." ECCV 2024
>
> [3] Liu, Yang, et al. "Citygaussian: Real-time high-quality large-scale scene rendering with gaussians." ECCV 2024
>
> [4] Chen, Yingshu, et al. "StyleCity: Large-Scale 3D Urban Scenes Stylization with Vision-and-Text Reference via Progressive Optimization." arXiv 2024
>
> [5]Brachmann, Eric, et al. "On the limits of pseudo ground truth in visual camera relocalization." ICCV 2021

---

> > ### Comment · Reviewer_HMwQ · 2024-11-20
> >
> > Thank you for your prompt response and for providing more extensive comparative experiments with HLoc, as well as the performance of GSLoc under different numbers of viewpoints. I still have some questions:
> > - On 7Scenes, does the 3DGS training use the poses from SfM GT or the poses obtained by running COLMAP separately? (i.e., Is the mapping process completely fair between GSLoc and HLoc?)
> > - Could you provide the corresponding results of HLoc using MASt3R for matching during localization?
> > - On 7Scenes, could you add the following images in the appendix to help understand the specific reasons why ACE + GSLoc significantly outperforms ACE + H-Loc:
> >     1. The query image
> >     2. The top-1 image obtained based on ACE pose + pose-based retrieval
> >     3. The rendered image used for GSLoc's pose estimation
> >     4. Additionally, it would be helpful to provide the following two results:
> > 	 + Visualization of the matching results between the query image and the top-1 retrieval image (showing SP+SG and MASt3R matches respectively would be great)
> > 	 + Visualization of the matching results between the query image and the GSLoc rendering (i.e., MASt3R matches)
> >
> > My concern is that if the query image and the nearest database image are nearly located, then H-Loc itself should be able to achieve good results. If the query image and the nearest database image are quite far, it would require 3DGS to have high-quality rendering under relatively large viewpoint changes, and the rendering need to be easy to match, to enable GSLoc to achieve results far surpassing HLoc. The above information can help me, and I believe,  relevant readers further understand GSLoc.

---

> ### Author Response · Authors · 2024-11-22
> **Response to Reviewer HMwQ**
>
> Thank you for your follow-up questions. We are happy to address your concerns.
>
> -(g) **Fair comparision**: On 7Scenes, we use the same SfM model for 3DGS training and HLoc relocalization with the same SfM GT. In the future, we will release all of our COLMAP models, pre-trained 3DGS models, and our code for reproducibility.
>
> -(h) **More visualization results**: In the new rebuttal version, we provide match visualizations of multiple scenes as well as examples of retrieved images in Figure 10 (page 18). We also give you detailed information about each image, which will help you reproduce the results easily. These examples illustrate the advantages of using 3DGS render images from SCR methods’ prediction to perform matching.  The key insight is that both image retrieval and ACE pose-based retrieval are restricted to identifying queries within a limited reference pool. In contrast, our approach theoretically allows for an unlimited reference pool.

---

> > ### Comment · Reviewer_HMwQ · 2024-11-22
> >
> > Thank you for your response. The additional information has deepened my understanding of the advantages of GSLoc / GS-CPR. By combining Scaffold-GS (w/ ACT) and MASt3R, even though the rendered results based on the initial poses estimated by ACE may contain some artifacts, this does not significantly affect the final matching performance of MASt3R or the improvement in localization accuracy. In addition, could you consider providing results of visual localization using MASt3R for matching within the H-Loc framework? This could be achieved by only leveraging the matches of already triangulated keypoints from the retrieved database images.

---

> > > ### Author Response · Authors · 2024-11-22
> > > **Response to Reviewer HMwQ**
> > >
> > > We appreciate your follow-up questions. We are pleased to present the results of the H-Loc experiments conducted on the 7Scenes dataset.  We leverage MASt3R to match the sparse keypoints of reference images from the SfM sparse point cloud.
> > >
> > > Average results across 7Scenes.
> > > | Methods | Avg. median error (cm/$^\circ$) $\downarrow$  |
> > > |-----------|----------|
> > > |NetVLAD + H-Loc (MASt3R) | 7.4/2.5|
> > > |ACE + H-Loc (MASt3R) |8.7/3.3|
> > > |ACE + GSLoc |**0.8/0.25**|

---

> > > > ### Comment · Reviewer_HMwQ · 2024-11-23
> > > >
> > > > Thank you for the response and the additional information. GSLoc is a simple yet promising pipeline, and I look forward to seeing its further development as its components continue to improve. I have raised my score accordingly.

---

### Official Review · Reviewer_rRXL · 2024-11-03

**Soundness:** 3
**Presentation:** 3
**Contribution:** 3
**Rating:** 6
**Confidence:** 3

**Summary:**

This method applies 3DGS to address the test-time camera pose refinement task. The manuscript first renders RGB and depth images from 3DGS, then matches the query image with the rendered images. To bridge the domain gap between the query image and 3DGS, an exposure-adaptive module is incorporated. Additionally, MASt3R is used as a training-free feature extractor for accurate matching. Extensive experiments demonstrate the effectiveness of the proposed GSLoc method.

**Strengths:**

S1 - The task appears meaningful, and the results presented in the manuscript are quite promising.

S2 - The Affine Color Transformation module is a noteworthy component, effectively addressing the RGB domain gap between the query image and the rendered images.

S3 - The experiments are well-designed and effectively demonstrate the superiority of the proposed GSLoc method over various baseline approaches.

**Weaknesses:**

W1 - The method appears to lack comparisons with state-of-the-art approaches, such as GLACE + GSLoc and ACE/GLACE + NeFes. Including these comparisons would provide a clearer benchmark for evaluating its performance.

W2 - I’m curious about the quality of the depth images rendered by 3DGS and whether they are consistent enough to serve as a reliable information source.

**Questions:**

Q1 - The sota comparision such as GLACE + GSLoc.

Q2 - The quality of the rendered depth image.

---

> ### Author Response · Authors · 2024-11-20
> **Response to Reviewer rRXL**
>
> Thank you very much for your positive feedback on our paper. We appreciate your comments on recognizing the novelty as well as the promising performance of our approach. We primarily address your concerns below.
>
> -(a) **W1**:  Here, we supplement the GLACE + GSLoc results, where GSLoc significantly improves GLACE accuracies in two of the three datasets (7scenes and 12scenes), demonstrating the effectiveness of our method. On indoor scenes, the initial accuracy of ACE and GLACE is very close.  The accuracy after refinement of GLACE is also very close to that of ACE with GSLoc refinement. On the Cambridge dataset, we achieve comparable results with an advantage on rotational error for GLACE.
>
> We conducted additional experiments with ACE+NeFeS and GLACE+NeFeS using the official open-source implementations. Our findings indicate that integrating NeFeS does not improve the accuracy of ACE or GLACE; in fact, it leads to a decline in performance. The quantitative results are presented below.
>
> **Table 1. Average results across 7Scenes.**
> | Method          | Avg. Median Err (cm/deg) ↓ | Avg. 5cm, 5deg (%) ↑ | Avg. 2cm, 2deg (%) ↑ |
> |------------------|---------------------|-----------------------|-----------------------|
> | ACE            | 1.1 / 0.34        | 97.1                 | 83.3                 |
> | ACE + GSLoc    | **0.8 / 0.25**         | **100**                 | **93.1**                 |
> |------------------|---------------------|-----------------------|-----------------------|
> | GLACE            | 1.2 / 0.36         | 95.6                 | 82.2                 |
> | GLACE + GSLoc    | **0.8 / 0.27**         | **99.5**                 | **90.7**                 |
>
> ---
>
> **Table 2. Average results across 12Scenes.**
> | Method          | Avg. Median Err (cm/deg) ↓ | Avg. 2cm, 2deg (%) ↑ |
> |------------------|---------------------|-----------------------|
> | ACE            | 0.7 / 0.26          | 97.2                 |
> | ACE + GSLoc    | **0.5 / 0.21**      | **98.7**                 |
> |------------------|---------------------|-----------------------|
> | GLACE            | 0.7 / 0.25          | 97.5                 |
> | GLACE + GSLoc    | **0.5 / 0.21**      | **98.9**                 |
>
> ---
>
>
> **Table3.  Average results across Cambridge Landmarks.**
> | Method           | Avg. Median Err (cm/deg) ↓ |
> |-------------------|---------------------|
> | ACE              | 21 / 0.47          |
> | ACE + NeFeS$_{50}$    | 22 / 0.56          |
> | ACE + GSLoc      | **17 / 0.33**          |
> |-------------------|---------------------|
> | GLACE            | **14** / 0.32          |
> | GLACE + NeFeS$_{50}$  | 21 / 0.50          |
> | GLACE + GSLoc    | **14 / 0.28**          |
>
> -(b) **W2**: Regarding the reliability of the depth map rendered from 3DGS, in our framework, we report the average Rel (Relative error) and $\delta_{1.25}$ (percentage of pixels where predicted depth lies within a factor of 1.25 of ground truth depth) across 17K test frames of 7Scenes using Kinect RGBD depth as GT.
>
> | Scene | Chess    | Fire     | Heads    | Office   | Pumpkin  | Kitchen  | Stairs   | avg. |
> |-----------|----------|----------|----------|----------|----------|----------|----------|----------|
> | Rel ($\downarrow$)| 0.16     | 0.12     | 0.11     | 0.22     | 0.46     | 0.32     | 1.32     | 0.40 |
> | $\delta_{1.25} (\uparrow)$| 85.7     | 89.5     | 86.9     | 76.1     | 85.3     | 83.7     | 79.8     |  83.9|
>
>
> The above table shows that depth maps rendered from 3DGS are accurate. For qualitative results, please refer to Figure 7 and Figure 8 in our paper. Since RANSAC can inherently provide robustness to some reasonable degree of noise, our evaluation results show that our 3DGS rendered depth maps can achieve “sufficient” robustness for high-accuracy pose prediction when combined with PnP+RANSAC. For example, we achieve 100% query accuracy of 5cm, 5deg on 7scenes datasets and nearly 99% accuracy on 2cm and 2deg on 12scenes datasets, demonstrating the reliability of our approach.

---

> > ### Comment · Reviewer_rRXL · 2024-11-25
> >
> > Thank you for your response. The additional information has successfully addressed the reviewer’s concerns. The reviewer acknowledges the significance of this work and believes it can contribute to the community.

---

### Official Review · Reviewer_BjDV · 2024-11-03

**Soundness:** 3
**Presentation:** 1
**Contribution:** 2
**Rating:** 8
**Confidence:** 4

**Summary:**

This paper presents a test-time refinement method, GSLoc, for improving camera localization accuracy. With a single RGB query, the method leverages ACT (exposure-adaptive affine color transformation) enhanced 3DGS and MASt3R to establish 2D-3D correspondences and then uses PnP + RANSAC for pose estimation. On two indoor and one outdoor datasets, the method achieved SOTA results.

**Strengths:**

++ The paper shows that, with introducing any new techniques, simply combining existing techniques can achieve state-of-the-art results for camera localization.

**Weaknesses:**

-- It appears that the authors misused \citep and \citet. This creates some difficulty when reading the paper.

-- Formulas should be followed by punctuation marks.

-- Please provide the detailed derivation of Eq. (3). It seems the translation part is incorrect.

-- Please provide analyses on the failure cases and potential limitations of the work.

-- The title is GSLoc, where GS stands for Gaussian Splatting, but the meaning of Loc is not mentioned in the paper. Since the focus of this work is on pose/localization refinement, Loc may not be the most suitable term.

**Questions:**

What are the failure cases and potential limitations of the work? Please provide some quantitative and qualitative results.

---

> ### Author Response · Authors · 2024-11-20
> **Response to Reviewer BjDV**
>
> Thank you for your positive comments, including the recognition that our proposed method is simple yet effective and achieves state-of-the-art results. Below, we address your concerns and respond to your questions.
>
> -(a) **Minor writing problem**: We modified the minor writing issues on ‘\cite’ and punctuation marks in the rebuttal version. Thank you for your valuable feedback on these writing details.
>
> -(b) **Eq. (3)**: Thanks for pointing out the typo here. The correct formula is:
>
> $$
> \hat{p}' = [\hat{R}' \mid \hat{t}'] = [R_{rel} \hat{R} \mid R_{rel}\hat{t} + st_{rel}]
> $$
>
>  Our code is implemented correctly. We appreciate the reviewer for catching this typo.
>
> -(c) **Limitation**: One limitation of our method lies in its dependency on the accuracy of the initial pose estimates provided by the pose estimator. When the initial pose is highly inaccurate, the overlap between the rendered images and the query image is insufficient to establish reliable 2D-3D correspondences for accurate pose estimation. Examples of failure cases are provided in Appendix A.5 of the rebuttal version. Following Section 4.5 of NeFeS [1], we conducted quantitative experiments to evaluate the limitations of our framework. Specifically, we introduced random perturbations to the ground truth poses of test frames on the ShopFacade scene, applying fixed magnitudes of rotational and translational errors independently. The results after pose refinement using GSLoc are presented below.
>
> **Table 1**:  Average rotation error after refinement by GSLoc
> | Jitter-magnitude ($^\circ$) | 5   | 10  | 20  | 30  | 40  | 50  | 55  | 60  |
> |-------------------------|------|------|------|------|------|------|------|------|
> | Avg. Rot. Error ($^\circ$) | 0.23 | 0.23 | 0.27 | 0.35 | 0.6  | 7    | 26   | 83   |
>
> **Table 2**: Average translation error after refinement by GSLoc
> | Jitter-magnitude (m)    | 1   | 2   | 3   | 4   | 5   | 6   |  8   | 10   |
> |-------------------------|------|------|------|------|------|------|------|------|
> | Avg. Trans. Error (m)  | 0.19 | 0.38 | 0.51 | 0.88 | 1.13 | 2.0   | 3.1 | 10.7 |
>
> Our framework can improve the accuracy when rotation < 50$^\circ$ and translation < 8 meters, respectively.  In contrast, NeFeS [1] achieves accuracy improvements only for rotational errors under 35$^\circ$ and translational errors up to 4 meters. These findings highlight that our method significantly expands the optimization range.
>
> -(d) **Title**: Thank you for your suggestion regarding the title. After careful discussion, we have decided to change the title and framework name from **GSLoc** to **GS-CPR** (**C**amera **P**ose **R**efinement) to better reflect our focus on camera pose refinement. We incorporated this change in the rebuttal version. We will keep using GSLoc in the remaining discussions with reviewers for clarity.
>
> [1] Chen, Shuai, et al. "Neural Refinement for Absolute Pose Regression with Feature Synthesis." CVPR. 2024.

---

> > ### Comment · Reviewer_BjDV · 2024-11-20
> >
> > Thanks for the clarifications. The rebuttal has addressed the reviewer's concerns. The reviewer thinks that such a simple and effective approach is beneficial to the community.

---

### Author Response · Authors · 2024-11-20
**General Response and Revision:**

Dear Reviewers,

We sincerely appreciate your recognition of the effectiveness of our approach and its state-of-the-art performance among neural render pose (NRP) estimation methods. Thank you for your thorough and insightful reviews, which have significantly contributed to improving our paper.

In this revision, we have implemented several changes in response to your valuable suggestions. These updates are highlighted in blue in the revised manuscript.

* Following Reviewer BjDV’s suggestion regarding the title, we carefully considered and updated the **title and framework name** from **GSLoc** (Localization) to **GS-CPR** (**C**amera **P**ose **R**efinement) to better reflect the focus of our work on camera pose/localization refinement. For clarity in discussions with reviewers, we continue to use the original name GSLoc, while the title change has been implemented in the rebuttal version. We also addressed the minor writing issues raised by Reviewer BjDV.

* Based on Reviewer UwQs’s feedback, we updated Table 4 to ensure consistent evaluation metrics across datasets.  We also included the configuration details of HLoc in Table 2, as recommended by Reviewer HMwQ.

* We added a new paragraph in Appendix A.3 to clarify our system design and included a limitation analysis in Appendix A.5. In addition, in response to Reviewer rRXL’s suggestion, we included the results of GLACE + our framework in Tables 12 and 13 of the Appendix A.6.

Finally, we have provided detailed responses to all your comments and concerns in the individual review threads. We hope that our revisions and responses address any remaining issues and help clarify the contributions of our work. We kindly request that you reconsider your evaluation of our paper in light of these changes.

---

### Meta-Review · Area_Chair_4Dt2 · 2024-12-21

**Metareview:**

This paper proposes a method for optimizing the initial camera pose in visual localization. Using 3DGS as the scene representation, RGB images and depth maps are rendered based on the initial camera pose of the query image estimated by other methods. Then, 2D-2D matches between the query image and the rendered results are computed via MASt3R and elevated to 2D-3D matches using the depth maps. Finally, the relative pose is solved through PnP to optimize the initial camera pose.  The idea of using 3DGS for visual localization is simple yet effective. Through high-quality 3DGS reconstruction and rendering, the proposed method directly computes optimized camera poses using image matching, which enhances the robustness and usability of the system.  On the other hand, the reviewers raised concerns regarding lacking certain experiments, insufficient analysis of the method’s applicability, incremental contribution, and limitation.  Some important SOTA methods are suggested for comparison.  The choice of the baseline is also argued. The combination of technologies is raised as a concern. Failure cases and/or clarifying potential limitations are also pointed out.  The authors have provided additional experiments to address the concerns on experiments in the rebuttal with arguing analysis of the results. The authors’ rebuttal and following discussion between the authors and the reviewers have resolved the raised concerns, leading to convincing contribution of the paper.  The reviewers are all positive for the paper.  This paper should be accepted, accordingly.  It is highly suggested for the authors to release the code in public so that the paper becomes more citable.

**Additional Comments On Reviewer Discussion:**

See above.

---

### Decision · Program_Chairs · 2025-01-22

Accept (Poster)